# NIRVANA: A SPECIALIZED GENERALIST MODEL WITH TASK-AWARE MEMORY MECHANISM

## ABSTRACT

Large Language Models (LLMs) have achieved remarkable success across a wide range of general language tasks but remain constrained in specialized domains. To address this problem, specialized memory mechanism can be used to enhance the model's ability on specialized tasks. Specialized Generalist Models (SGMs) aim to preserve broad capabilities while achieving expert-level performance in target domains via test-time task identification and reconfiguration. However, traditional LLM structures including Transformer, Linear Attention, and hybrid models do not employ specialized memory mechanism guided by task information. In this paper, we present Nirvana, an SGM with specialized memory mechanism, linear time complexity, and test-time task information extraction. Besides, we propose the Task-Aware Memory Trigger (*Trigger*) that flexibly adjusts memory mechanism based on the current task's requirements. In Trigger, each incoming sample is treated as a self-supervised fine-tuning task, enabling Nirvana to adapt its task-related parameters on the fly to domain shifts. We also design the Specialized Memory Updater (*Updater*) that dynamically memorizes the context guided by Trigger. We conduct experiments on both general language tasks and multiple specialized domains. Nirvana matches or exceeds the performance of LLM baselines on general benchmarks, while achieving the lowest perplexity across specialized domains including biomedicine, finance, and law. On the challenging task of Magnetic Resonance Imaging (MRI), we attach lightweight codecs to the frozen Nirvana backbone and fine-tune them on paired k-space measurements and images. Trigger enables effective adaptation to the MRI domain by adjusting task-related parameters during inference, even without updating the backbone. Nirvana yields higher-fidelity MRI reconstructions than conventional MRI models and LLM-based models, and it also generates reliable preliminary clinical reports. Ablation studies show that removing Trigger results in notable performance degradation across all evaluation tasks, demonstrating its essential role in task-aware specialization.

## 1 INTRODUCTION

Large Language Models (LLMs) have significantly advanced general language processing, but still have limitations in specialized tasks (Jimenez et al., 2023; Guha et al., 2023; Srivastava et al., 2022; Liang et al., 2022). For instance, while an LLM can describe the rules of the game of Go, it struggles to match the deep, domain-specific strategic reasoning of expert Go programs like AlphaGo. To solve this problem, Specialized Generalist Models (SGMs) (Zhang et al., 2024) are proposed to retain broad, generalist capabilities while achieving expert-level performance in at least one (and ideally multiple) specialized domains. SGMs play a pivotal role in real deployments, e.g., medicine and other safety-critical workflows, which demand both general reasoning ability and domain-expert inference accuracy, together with verifiable use of external knowledge and tools (Wang et al., 2025; Lewis et al., 2020; Schick et al., 2023; Yao et al., 2023).

Specifically, the specialized memory mechanism of SGMs requires that models can identify the task information on the fly and then adapt their internal pathways and memory use, explicit retrieval and non-parametric memory (Fedus et al., 2021; Lepikhin et al., 2020; Jiang et al., 2024; Lewis et al., 2020; Borgeaud et al., 2022; Khandelwal et al., 2019; Wu et al., 2022), as well as the ability to dynamically choose the methodology to memorize. Diverse memory mechanisms have been explored to capture, store, and adapt contextual information as sequences grow longer, which is summarized in

| Model | Dynamic Decay | Non-Linearity | Local Optimum | Specialized Memory | Memory Update Operation |
|---|---|---|---|---|---|
| Attention | ✗ | ✓ | ✓ | ✗ | $M_t = M_{t-1} \cup \{(\boldsymbol{k}_t, \boldsymbol{v}_t)\}$ |
| SWA | ✗ | ✓ | ✓ | ✗ | $M_t = (M_{t-1} \setminus \{(\boldsymbol{k}_c, \boldsymbol{v}_c)\}) \cup \{(\boldsymbol{k}_t, \boldsymbol{v}_t)\}$ |
| Naive Linear Attention | ✗ | ✗ | ✗ | ✗ | $M_t = M_{t-1} + \boldsymbol{v}_t \boldsymbol{k}_t^\top$ |
| DeltaNet | ✗ | ✗ | ✗ | ✗ | $M_t = \left(I - \beta_t \boldsymbol{k}_t \boldsymbol{k}_t^\top\right) M_{t-1} + \beta_t \boldsymbol{v}_t \boldsymbol{k}_t^\top$ |
| Longhorn | ✗ | ✗ | ✗ | ✗ | $M_t = \left(I - \delta_t \boldsymbol{k}_t \boldsymbol{k}_t^\top\right) M_{t-1} + \delta_t \boldsymbol{v}_t \boldsymbol{k}_t^\top$ |
| RetNet/Lightning | ✗ | ✗ | ✗ | ✗ | $M_t = \alpha M_{t-1} + \boldsymbol{v}_t \boldsymbol{k}_t^\top$ |
| GLA | ✓ | ✗ | ✗ | ✗ | $M_t = \mathrm{Diag}\left(\alpha_t\right) M_{t-1} + \boldsymbol{v}_t \boldsymbol{k}_t^\top$ |
| HGRN2 | ✓ | ✗ | ✗ | ✗ | $M_t = \mathrm{Diag}\left(\boldsymbol{a}_t\right) M_{t-1} + \boldsymbol{v}_t (\mathbf{1} - \boldsymbol{a}_t)^\top$ |
| Mamba2 | ✓ | ✗ | ✗ | ✗ | $M_t = \alpha_t M_{t-1} + \beta_t \boldsymbol{v}_t \boldsymbol{k}_t^\top$ |
| PolySketchFormer | ✗ | ✓ | ✗ | ✗ | $M_t = M_{t-1} + \boldsymbol{v}_t \left(\boldsymbol{k}_t^\top\right)^p$ |
| TTT | ✗ | ✓ | ✗ | ✗ | $M_t = M_{t-1} - \eta_t \nabla \ell \left(M_{t-1}(\boldsymbol{k}_t), \boldsymbol{v}_t\right)$ |
| RWKV-7 | ✓ | ✗ | ✗ | ✗ | $M_t = \left(\mathrm{Diag}\left(\alpha_t\right) - \beta_t \boldsymbol{k}_t \boldsymbol{k}_t^\top\right) M_{t-1} + \beta_t \boldsymbol{v}_t \boldsymbol{k}_t^\top$ |
| Gated DeltaNet | ✓ | ✗ | ✗ | ✗ | $M_t = \alpha_t \left(I - \beta_t \boldsymbol{k}_t \boldsymbol{k}_t^\top\right) M_{t-1} + \beta_t \boldsymbol{v}_t \boldsymbol{k}_t^\top$ |
| Titans | ✓ | ✓ | ✗ | ✗ | $M_t = \alpha_t M_{t-1} + S_t$ 
 $S_t = \eta_t S_{t-1} - \eta_t \nabla \ell \left(M_{t-1}(\boldsymbol{k}_t), \boldsymbol{v}_t\right)$ |
| Nirvana | ✓ | ✓ | ✓ | ✓ | $M_t = \gamma_t \{\alpha_t (I - \beta_t \boldsymbol{k}_t \boldsymbol{k}_t^\top) M_{t-1}^{\mathrm{LA}} + \beta_t \boldsymbol{v}_t \boldsymbol{k}_t^\top\}$ 
 $\cup \eta_t \{(M_{t-1}^{\mathrm{SWA}} \setminus \{(\boldsymbol{k}_c, \boldsymbol{v}_c)\}) \cup (\boldsymbol{k}_t, \boldsymbol{v}_t)\}$ |

Table 1: A summary of some modern LLM architectures. We compare them based on 4 characteristics: Dynamic Decay: adaptively forget memory about the past; Non-Linearity: beyond linear algebra operations such as matrix multiplications; Local Optimum: extract the second-order information about tokens; Specialized Memory: adaptively memorize the context according to the task information. Subscript $t$ refers to the token's position; $\alpha_t$ is token-dependent, while $\alpha$ is token-independent.

Table 1. However, prevailing LLM architectures still fall short on flexible and specialized memory (Behrouz et al., 2024; Yang et al., 2025).

Transformer's self-attention (Vaswani et al., 2017) models all past tokens in parallel, offering strong expressivity but at quadratic cost. Sliding Window Attention (SWA) (Fu et al., 2025) reduces resources by keeping only a fixed buffer of key-value pairs, while Linear Attention scales linearly by compressing history into a memory matrix. Naive Linear Attention (Katharopoulos et al., 2020) does this via outer-product updates, refined by RetNet (Sun et al., 2023) with decayed retention factors. Related variants include HGRN1/2 (Qin et al., 2024), Mamba1/2 (Gu & Dao, 2023; Dao & Gu, 2024), and RWKV6/7 (Peng et al., 2023; 2024; 2025), which use data-dependent decay. Delta-rule updates (Yang et al., 2024) subtract aligned past memory before adding new content, extended by Gated DeltaNet (Yang et al., 2025) with learnable gating. Test-Time Training (TTT) (Sun et al., 2024) instead treats hidden-state updates as self-supervised adaptation, while Titans (Behrouz et al., 2024) introduce deep MLP-based memory with momentum updates. Hybrid models blend these ideas: Samba (Ren et al., 2024) interleaves Mamba with SWA, Jamba (Lieber et al., 2024) mixes Transformer and Mamba under MoE gating, and Gated DeltaNets (H1/H2) (Yang et al., 2025) combine SWA, Mamba, and Gated DeltaNet. These hybrid models yield improved length-extrapolation and reasoning capabilities over pure Linear Attention. However, all the above LLM architectures still fail to answer the question: **How to flexibly change the memory mechanism in a specialized way according to the task during the test time?**

To answer this question, we propose a novel SGM called Nirvana, which realizes specialized and non-linear memory mechanism with dynamic decay and second-order information. We propose a Task-Aware Memory Trigger (*Trigger*), which enables dynamic self-supervised fine-tuning to adapt to domain shifts. By turning each incoming sample into a learning task, Trigger continuously refines the model's fast parameters on the fly, boosting robustness under varying data conditions. We also design a Specialized Memory Updater (*Updater*) that dynamically memorizes the context under the guidance of Trigger. In experiments, Nirvana matches or surpasses strong LLM baselines on standard general-language benchmarks, while achieving the lowest perplexity across specialized domains including biomedicine, finance, and law. On the challenging task of Magnetic Resonance Imaging (MRI), we attach lightweight codecs to the frozen Nirvana backbone and fine-tune them on paired k-space signals and images, achieving higher-fidelity reconstructions than conventional LLM-based models. In this setting, Trigger enables Nirvana to calibrate itself to the distribution of k-space signals and MRI images, yielding diagnostic-quality reconstructions and accurate clinical reports. This unified approach of Trigger and Updater obviates the need for extensive domain-specific model backbone retraining, saving valuable time and data resources. By seamlessly fusing broad linguistic

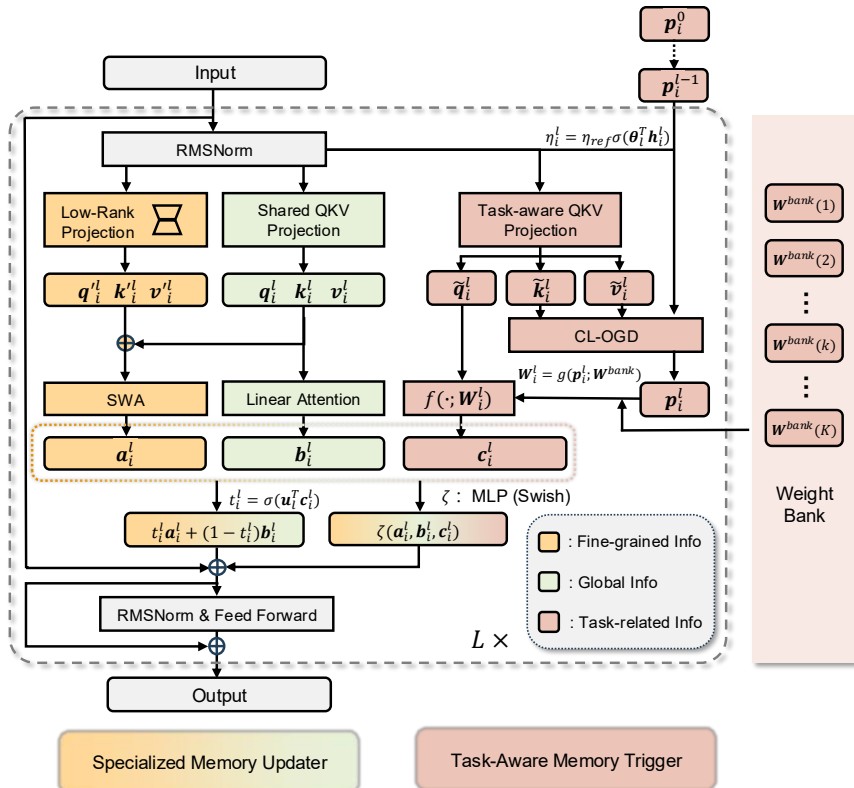

Figure 1: Visualization of Nirvana's architecture. Updater employs conditional interpolation between SWA and Linear Attention. Trigger extracts fast parameters $\boldsymbol{p}_i^l$ to update $f(\cdot; \boldsymbol{W}_i^l)$ and generates task-related information $\boldsymbol{c}_i^l$ as the condition of Updater. We can use an arbitrary architecture for Linear Attention from the 3-rd to the last but not least line in Table 1.

intelligence with rapid, on-the-fly specialization, Nirvana ushers in a new class of general-to-special SGMs.

## 2 METHOD

In order to realize the specialized memory mechanism, we propose two branches corresponding to two levels of memory. The first branch Trigger is designed to memorize the abstract task information, and the second branch Updater is designed to adaptively memorize the detailed context according to the task information. The two branches interact in a cross-layer manner, where Trigger provides the task information as conditions to Updater for interpolation between SWA and Linear Attention. The architecture of Nirvana is shown in Figure 1.

### 2.1 SPECIALIZED MEMORY UPDATER

SWA excels at modeling fine-grained information and local dependencies within a bounded context (Vaswani et al., 2017), while Linear Attention enables accurate global information modeling of long sequences (Yang et al., 2025). Therefore, Updater is proposed to combine the advantages of both modules. We employ SWA instead of the full attention, such that the computational complexity of the model only grows *linearly* with the length of the input sequence. With the aim of sharing the majority of parameters across the network, we use shared QKV projection matrices for SWA and Linear Attention. In order to learn the discrepancy of query, key, and value between the two modules with small computation overhead and few learnable parameters, we use a dimension reduction representation with low rank linear projection independently added before SWA. Denote $\boldsymbol{q}_i'^l$, $\boldsymbol{k}_i'^l$, and

$\boldsymbol{v}'^l_i$ as the outputs of low rank linear projection, where the superscript $l$ and the subscript $i$ denote the $l$-th layer and the $i$-th token throughout this paper. We add $\boldsymbol{q}'^l_i$, $\boldsymbol{k}'^l_i$, and $\boldsymbol{v}'^l_i$ to the original counterparts to yield the query, key, and value of SWA.

## 2.2 Task-Aware Memory Trigger

The process of memory-related parametric learning can be viewed as compressing a massive training set into the weights of a model. This process of compression into the weights involves capturing the essence of the data that the model has been trained on. In conventional frameworks, the model's weights are shared across different tokens. However, Nirvana introduces a novel methodology that tailors the fast weights specifically for different layers and different tokens through Trigger.

In order to facilitate the continuous flow of task-related information across various layers, we propose Trigger, which updates the fast weights by extracting the task in the context. Besides, we design a novel mechanism that allows the tokens not to share the same fast weights, such that the model can avoid information leakage during the training process. Specifically, the tokens have individual fast hyper-parameters $\boldsymbol{p}^l_i$ that are implicitly determined by the task information. In order to map $\boldsymbol{p}^l_i$ to the fast weights of a neural network, we extract the tokens' individual fast weights from a fast weight bank $\boldsymbol{W}^{\text{bank}}$, which is shared across different layers and different tokens. Generally, the fast weights are denoted by $\boldsymbol{W}^l_i$ and are generated given $\boldsymbol{p}^l_i$ and $\boldsymbol{W}^{\text{bank}}$ through a predefined function as

$$\boldsymbol{W}^l_i = g(\boldsymbol{p}^l_i; \boldsymbol{W}^{\text{bank}}). \tag{1}$$

Note that the generation of fast weights is conditional on both the layer and the token, allowing for a more granular control of the learning process across the network. Besides, the dimension of $\boldsymbol{p}^l_i$ should be much lower than $\boldsymbol{W}^l_i$, which ensures efficient parameter transfer across layers.

To extract the abstract task-related information, we employ linear layers to compute the query, key, and value of Trigger, denoted by $\tilde{\boldsymbol{q}}^l_i$, $\tilde{\boldsymbol{k}}^l_i$, and $\tilde{\boldsymbol{v}}^l_i$, respectively. For computational efficiency, $\tilde{\boldsymbol{q}}^l_i$, $\tilde{\boldsymbol{k}}^l_i$, and $\tilde{\boldsymbol{v}}^l_i$ have a relatively low dimension compared to the dimension of hidden states. The task extraction process can be formulated as

$$\boldsymbol{c}^l_i = f(\tilde{\boldsymbol{q}}^l_i; \boldsymbol{W}^l_i), \tag{2}$$

where $f(\tilde{\boldsymbol{q}}^l_i; \boldsymbol{W}^l_i)$ is a meta function modeled by a neural network that takes $\tilde{\boldsymbol{q}}^l_i$ as input and uses $\boldsymbol{W}^l_i$ as the network's test-time-changeable parameters. In order to update the fast parameters $\boldsymbol{p}^l_i$, we propose the Cross-Layer Online Gradient Descent (CL-OGD) algorithm. CL-OGD guides Nirvana to update $f(\cdot; \boldsymbol{W}^l_i)$ at the test time by minimizing the following loss function:

$$\mathcal{L}^l_i = \|f(\tilde{\boldsymbol{k}}^l_i; \boldsymbol{W}^l_i) - \tilde{\boldsymbol{v}}^l_i\|^2_2. \tag{3}$$

Since the parameters of $f(\cdot; \boldsymbol{W}^l_i)$ are decided by $\boldsymbol{p}^l_i$ according to (1), updating $f(\cdot; \boldsymbol{W}^l_i)$ in CL-OGD is equivalent to updating the fast parameters $\boldsymbol{p}^l_i$ in the test time, which can be formulated as

$$\Delta \boldsymbol{p}^l_i = \frac{\partial \mathcal{L}^l_i}{\partial \boldsymbol{W}^l_i} \frac{\partial \boldsymbol{W}^l_i}{\partial \boldsymbol{p}^l_i} = \frac{\partial \|f(\tilde{\boldsymbol{k}}^l_i; \boldsymbol{W}^l_i) - \tilde{\boldsymbol{v}}^l_i\|^2_2}{\partial \boldsymbol{W}^l_i} \frac{\partial g(\boldsymbol{p}^l_i; \boldsymbol{W}^{\text{bank}})}{\partial \boldsymbol{p}^l_i}, \tag{4}$$

$$\boldsymbol{p}^l_i = \boldsymbol{p}^{l-1}_i - \eta^l_i \Delta \boldsymbol{p}^l_i, \tag{5}$$

where $\eta^l_i$ is the adaptive online learning rate, defined as $\eta^l_i = \eta_{\text{ref}} \sigma(\boldsymbol{\theta}^\top_l \boldsymbol{h}^l_i)$ with $\boldsymbol{\theta}_l$ being a learnable projection vector and $\boldsymbol{h}^l_i$ the hidden state before QKV projection. The function $\sigma(\cdot)$ denotes the Sigmoid function, and $\eta_{\text{ref}}$ is a reference learning rate. In Nirvana, task-relevant information in the hidden states is extracted only after several prelude layers. Accordingly, in the first post-prelude layer, $\boldsymbol{p}^0_i$ is initialized as an all-ones vector for every token.

Since the hidden states vary in magnitude in different layers, we use Layer Normalization (LN) in $f(\boldsymbol{x}; \boldsymbol{W})$ for better stability. To realize the relatively low computational complexity, $f(\boldsymbol{x}; \boldsymbol{W})$ contains a linear layer, an LN operation, and a residual connection, i.e., $f(\boldsymbol{x}; \boldsymbol{W}) = \boldsymbol{x} + \text{LN}(f_{\text{linear}}(\boldsymbol{x}; \boldsymbol{W})) = \boldsymbol{x} + \text{LN}(\boldsymbol{W}_{\text{linear}} \boldsymbol{x} + \boldsymbol{b}_{\text{linear}})$. For the simplicity of calculation and for the ease of back propagation, we design a weight sharing mechanism that allows for the reuse of the same weight across different layers and tokens. The weight sharing mechanism is implemented through a

learnable weight bank $\boldsymbol{W}^{\text{bank}}$ that stores the shared weight parameters across different layers and tokens. The function $g(\boldsymbol{p}_i^l; \boldsymbol{W}^{\text{bank}})$ is formulated as:

$$g(\boldsymbol{p}_i^l; \boldsymbol{W}^{\text{bank}}) = \sum_{k=1}^{K} \boldsymbol{p}_i^l(k) \boldsymbol{W}^{\text{bank}}(k), \tag{6}$$

where $\boldsymbol{p}_i^l(k)$ denotes the $k$-th element of $\boldsymbol{p}_i^l$ and $\boldsymbol{W}^{\text{bank}}(k)$ denotes the $k$-th block of $\boldsymbol{W}^{\text{bank}}$, respectively. Compared to the hidden states $\boldsymbol{h}_i^l$, $\boldsymbol{p}_i^l$ has a much smaller dimension (e.g., $K = 64$), enabling efficient parameter transfer across layers. Besides, $\boldsymbol{p}_i^l$ is updated according to Equation 4 with the gradient computed as:

$$\frac{\partial g(\boldsymbol{p}_i^l; \boldsymbol{W}^{\text{bank}})}{\partial \boldsymbol{p}_i^l} = [\text{vec}\{\boldsymbol{W}^{\text{bank}}(1)\}, \ldots, \text{vec}\{\boldsymbol{W}^{\text{bank}}(K)\}]. \tag{7}$$

Since task information is inherently high-level and difficult to extract within the early layers of Nirvana, we designate the first $N_{\text{pre}}$ layers as prelude layers, which operate without the Trigger and are not involved in task-information extraction. These prelude layers use only Linear Attention, while SWA is introduced in the subsequent post-prelude layers. Because standard Linear Attention architectures (e.g., Gated DeltaNet, Mamba2) are already capable of capturing position-dependent structure in the input sequence (Yang et al., 2025), there is no need to apply Rotary Positional Embedding (RoPE) (Su et al., 2024) before SWA. Incorporating RoPE at this stage would introduce unnecessary computation and could weaken the model's ability to extrapolate to context lengths beyond those seen during training. Additional experiments and comparisons regarding RoPE are provided in Appendix A.4.

The outputs of the SWA and the Linear Attention module are integrated by a conditional interpolation mechanism. Let the output of the SWA module be denoted by $\boldsymbol{a}_i^l$ and the output of the Linear Attention module be denoted by $\boldsymbol{b}_i^l$. The conditional interpolation mechanism is defined as

$$v_i^l = t_i^l \boldsymbol{a}_i^l + (1 - t_i^l) \boldsymbol{b}_i^l + \zeta(\boldsymbol{a}_i^l, \boldsymbol{b}_i^l, \boldsymbol{c}_i^l), \tag{8}$$

where $t_i^l \in (0, 1)$ is a task-dependent scalar that controls the interpolation between the outputs of the two modules. Specifically, $t_i^l$ is computed as $t_i^l = \sigma(\boldsymbol{u}_l^\top \boldsymbol{c}_i^l)$, where $\boldsymbol{u}_l$ is a learnable projection column vector. Besides, $\zeta(\boldsymbol{a}_i^l, \boldsymbol{b}_i^l, \boldsymbol{c}_i^l)$ adds a non-linear supplement to the conditional interpolation. Specifically, $\zeta(\boldsymbol{a}_i^l, \boldsymbol{b}_i^l, \boldsymbol{c}_i^l)$ is a two-layer MLP with Swish activation function, and maps the concatenation of $\boldsymbol{a}_i^l, \boldsymbol{b}_i^l, \boldsymbol{c}_i^l$ to a vector at the same length of $\boldsymbol{a}_i^l$. In order to make the number of the parameters in $\zeta(\boldsymbol{a}_i^l, \boldsymbol{b}_i^l, \boldsymbol{c}_i^l)$ relatively small, the length of the hidden layer in $\zeta(\boldsymbol{a}_i^l, \boldsymbol{b}_i^l, \boldsymbol{c}_i^l)$ is $1/8$ of the length of $\boldsymbol{a}_i^l$. The output of the conditional interpolation module $v_i^l$ is then sent to the following RMSNorm and Feed-Forward Network (FFN) as the input.

## 3 EXPERIMENTS

In experiments, we employ Gated DeltaNet (Yang et al., 2025) in the linear attention part of Nirvana, due to the outstanding performance of Gated DeltaNet in language modeling tasks. We train Nirvana from scratch with a training context window of length 4096 and a global batch size of 0.5M tokens. We use a model size of 1.3B parameters and train the model on 100B tokens sampled from the FineWeb dataset (Penedo et al., 2024). The window length of SWA in Updater is set as 2048. We employ the AdamW optimizer (Loshchilov et al., 2017) and a hybrid learning rate schedule of linear warm-up (the first 1B tokens) followed by the cosine decay, reaching a peak learning rate of $4 \times 10^{-4}$. We utilize the LLaMA-2 tokenizer with a vocabulary size of 32,000. Training is conducted on 64 NVIDIA A800 GPUs. In evaluation, perplexity (ppl), accuracy (acc), and normalized accuracy (acc_n) are measured with held-out test data on 8 NVIDIA A800 GPUs. We also conduct the ablation study, where Nirvana-noTrigger refers to the Nirvana model without Trigger extracting the task-related information.

### 3.1 GENERAL LANGUAGE MODELING

#### 3.1.1 PERFORMANCE COMPARISON

In Table 2, we report the models' language modeling performance using ppl on 2 datasets: Wikitext (Wiki.) and LAMBADA (LMB.), and we also evaluate the models' zero-shot common sense reasoning

Table 2: Language modeling and zero-shot common sense reasoning performance of 1.3B models.

| Model | Wiki. ppl ↓ | LMB. ppl ↓ | LMB. acc ↑ | PIQA acc ↑ | Hella. acc_n ↑ | Wino. acc ↑ | ARC-e acc ↑ | ARC-c acc_n ↑ | SIQA acc ↑ | BoolQ acc ↑ | Avg. ↑ |
|---|---|---|---|---|---|---|---|---|---|---|---|
| Transformer++ | 18.53 | 18.32 | 42.60 | 70.02 | 50.23 | 53.51 | 68.83 | 35.10 | 40.66 | 57.09 | 52.25 |
| RetNet | 19.08 | 17.27 | 40.52 | 70.07 | 49.16 | 54.14 | 67.34 | 33.78 | 40.78 | 60.39 | 52.02 |
| HGRN2 | 19.10 | 17.69 | 39.54 | 70.45 | 49.53 | 52.80 | 69.40 | 35.32 | 40.63 | 56.66 | 51.79 |
| Mamba | 17.92 | 15.06 | 43.98 | 71.32 | 52.91 | 52.95 | 69.52 | 35.40 | 37.76 | 61.13 | 53.12 |
| Mamba2 | 16.56 | 12.56 | 45.66 | 71.87 | 55.67 | 55.24 | **72.47** | 37.88 | 40.20 | 60.13 | 54.89 |
| DeltaNet | 17.71 | 16.88 | 42.46 | 70.72 | 50.93 | 53.35 | 68.47 | 35.66 | 40.22 | 55.29 | 52.14 |
| Gated DeltaNet | 16.42 | 12.17 | 46.65 | 72.25 | 55.76 | 57.45 | 71.21 | 38.39 | 40.63 | 60.24 | 55.32 |
| Samba | 16.13 | 13.29 | 44.94 | 70.94 | 53.42 | 55.56 | 68.81 | 36.17 | 39.96 | 62.11 | 54.00 |
| Gated DeltaNet-H1 | 16.07 | 12.12 | 47.73 | 72.57 | 56.53 | 58.40 | 71.75 | **40.10** | 41.40 | **63.21** | 56.40 |
| Gated DeltaNet-H2 | **15.91** | 12.55 | 48.76 | 72.19 | 56.88 | 57.77 | 71.33 | 39.07 | **41.91** | 61.55 | 56.18 |
| Nirvana-noTrigger | 16.60 | 12.25 | 49.40 | 73.12 | 57.43 | 59.27 | 68.80 | 37.84 | 41.50 | 54.68 | 55.26 |
| **Nirvana (Ours)** | 16.05 | **11.56** | **50.37** | **73.67** | **58.25** | **59.48** | 69.92 | 39.51 | 41.62 | 59.27 | **56.51** |

performance using acc and acc_n on 8 datasets: LAMBADA (LMB.), PIQA, HellaSwag (Hella.), WinoGrande (Wino.), ARC-easy (ARC-e), ARC-challenge (ARC-c), SIQA, and BoolQ. On Wiki. dataset, Nirvana achieves a ppl of 16.05, which is slightly higher than the SOTA model (15.91, Gated DeltaNet-H2). Notably, on LMB. dataset, Nirvana achieves a ppl of 11.56, which is better than the SOTA model (12.12, Gated DeltaNet-H1). On common sense reasoning tasks, Nirvana outperforms all the other models and achieves the highest accuracy on LMB., PIQA, Hella., and Wino. datasets. The performances of Nirvana on ARC-e, ARC-c, SIQA, and BoolQ are slightly worse than the SOTA models, but are still comparable. Moreover, Nirvana achieves the highest average accuracy on common sense reasoning tasks. In ablation study, the performance of Nirvana is better than Nirvana-noTrigger in terms of the average accuracy.

Table 3: S-NIAH performance of 1.3B models. S-NIAH-PK, S-NIAH-N, and S-NIAH-W are 3 tasks for single pass-key retrieval in a haystack, single number in a haystack, and single word in a haystack, respectively. All models are trained with 4K context length.

| Model | S-NIAH-PK | | | S-NIAH-N | | | S-NIAH-W | | | Avg. |
|---|---|---|---|---|---|---|---|---|---|---|
| | 2K | 4K | 8K | 2K | 4K | 8K | 1K | 2K | 4K | |
| Transformer++ | **100.0** | **100.0** | 62.6 | **100.0** | **100.0** | 59.4 | **100.0** | **100.0** | **98.6** | 91.2 |
| Mamba2 | 98.6 | 61.4 | 31.0 | 98.4 | 55.8 | 14.2 | 62.2 | 42.2 | 4.2 | 52.0 |
| DeltaNet | 96.8 | 98.8 | 98.6 | 47.2 | 15.4 | 12.8 | 85.2 | 46.2 | 20.0 | 57.9 |
| Gated DeltaNet | 89.8 | 91.4 | 90.0 | 99.2 | 91.8 | 26.4 | 86.4 | 82.6 | 24.4 | 75.8 |
| TTT | 98.4 | 98.8 | 98.0 | 60.2 | 36.6 | 10.2 | 85.8 | 78.8 | 28.0 | 66.1 |
| Samba | 98.8 | 98.0 | 97.4 | 98.8 | 98.6 | 96.2 | 97.4 | 96.8 | 90.0 | 96.9 |
| Gated DeltaNet-H2 | 99.2 | 97.8 | 97.4 | 98.0 | 97.8 | 96.2 | 98.0 | 97.4 | 96.8 | 97.6 |
| Nirvana-noTrigger | 99.6 | 99.6 | 99.0 | 99.8 | 99.8 | 98.8 | 99.0 | 97.4 | 94.8 | 98.6 |
| **Nirvana (Ours)** | **100.0** | **100.0** | **100.0** | **100.0** | **100.0** | **99.6** | 98.8 | 97.8 | 95.4 | **99.1** |

We evaluate Nirvana on Single Needle-In-A-Haystack (S-NIAH) benchmark with different context lengths according to RULER (Hsieh et al., 2024). In Table 3, Nirvana outperforms all the other models in S-NIAH-PK and S-NIAH-N, and achieves the highest average accuracy. Particularly, in S-NIAH-PK (2K, 4K, and 8K context length) and in S-NIAH-N (2K and 4K context length), Nirvana achieves 100% accuracy, remarkably higher than most of the existing models. When trained with 4K context length and tested with 8K context length, Transformer++ does not perform well due to its relatively poor extrapolation ability. However, Nirvana maintains its superior performance with 8K context length, which illustrates its solid extrapolation capability. Notably, Nirvana-noTrigger performs worse than Nirvana, but is still better than other models on the average accuracy.

Table 4: Inference speed of 1.3B models at prompt sequence length 4096, with batchsize = 4.

| Model | Inference Speed (tokens/s) |
|---|---|
| Transformer++ | 191 |
| Mamba2 | 413 |
| Gated DeltaNet | 461 |
| Samba | 497 |
| Nirvana-noTrigger | **568** |
| Nirvana | 516 |

Table 5: Perplexity of 1.3B models on three specialized domains. Lower is better.

| Model | Biomedicine | Finance | Law | Avg. |
|---|---|---|---|---|
| Transformer++ | 9.28 | 10.70 | 8.82 | 9.60 |
| Mamba2 | 9.13 | 9.97 | 9.07 | 9.39 |
| Gated DeltaNet | 9.02 | 9.72 | 8.89 | 9.21 |
| Samba | 9.27 | 9.50 | 8.74 | 9.17 |
| Nirvana-noTrigger | 9.19 | 9.87 | 8.84 | 9.30 |
| **Nirvana** | **8.25** | **7.88** | **7.22** | **7.78** |

### 3.1.2 INFERENCE EFFICIENCY

To quantify the computational complexity, we report the inference speed of all 1.3B models at a prompt length of 4096 tokens in Table 4. Among all non-ablated models, Nirvana achieves the highest inference speed at 516 tokens/s, outperforming Samba (497 tokens/s), Gated DeltaNet (461 tokens/s), and Mamba2 (413 tokens/s), while substantially exceeding the classical Transformer++ baseline (191 tokens/s). Nirvana-noTrigger achieves the highest inference speed at 568 tokens/s, but its slight advantage over full Nirvana comes with a substantial performance drop consistently observed across the earlier and later ablation experiments, clearly demonstrating the essential role of Trigger. Since Trigger operates on a compact 64-dimensional space, it adds only a negligible linear-time cost. Moreover, Updater's SWA and Linear Attention modules maintain linear complexity, yielding faster inference than both Mamba2 and full-attention baselines. When combined with the accuracy gains in other experiments, these findings indicate that Nirvana improves model performance while simultaneously offering the best inference efficiency among the evaluated baselines.

### 3.2 SPECIALIZED ABILITY EVALUATION

### 3.2.1 PERFORMANCE IN SPECIALIZED DOMAINS: BIOMEDICINE, FINANCE, AND LAW

In order to assess performance in specialized domains, we evaluate various 1.3B models, including the ablated Nirvana-noTrigger, on three specialized corpora: (1) biomedical text from MIMIC-III clinical notes (Johnson et al., 2016), (2) financial news from April 2024 to October 2024 utilized in FinGPT (Liu et al., 2023), and (3) legal documents from the Asylex refugee-status corpus (Barale et al., 2023). All models are fine-tuned for 3 epochs on each domain. As shown in Table 5, Nirvana achieves the lowest perplexity in every domain and the best overall average of 7.78, substantially outperforming Transformer++, Mamba2, Gated DeltaNet, and Samba, whose averages range from 9.17 to 9.60. Importantly, the ablated Nirvana-noTrigger performs similarly to strong baselines but remains noticeably weaker than full Nirvana across all three domains, with an average perplexity gap of over 1.5 points. This consistent discrepancy highlights the central role of Trigger in enabling Nirvana to adapt effectively to specialized-domain distributions, demonstrating that Trigger materially enhances domain-specific modeling beyond what the backbone alone can achieve.

### 3.2.2 MRI RECONSTRUCTION AND REPORT GENERATION

MRI reconstruction is a critical medical process aimed at enhancing MRI image quality while reducing the scanning time, which is both clinically significant and technically demanding. Thus,

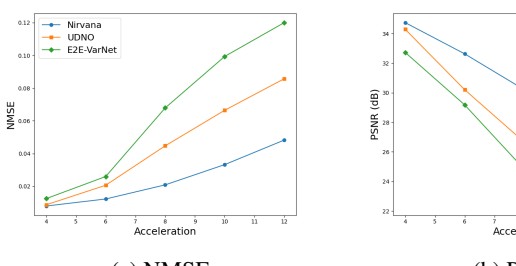 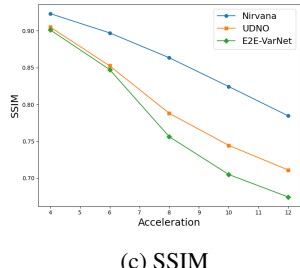

|  (a) NMSE | (b) PSNR | (c) SSIM |

Figure 2: MRI reconstruction performance comparison for models with 160M trainable parameters. The acceleration rate is also the undersampling rate.

it serves as a rigorous testbed to evaluate Nirvana's specialized ability. We explore the specialized application of Nirvana on the task of MRI reconstruction. In this task, Nirvana takes the raw k-space signals collected by the coils and an instruction prompt as input, and outputs the reconstructed image tokens and the corresponding analysis tokens of the reconstructed image. In order to transform the k-space signals into the embedding space of Nirvana, we use a multi-coil Variational Network (VarNet) (Giannakopoulos et al., 2024) followed by a lightweight ViT network (Yuan et al., 2021), which is defined as the k-space encoder. The k-space encoder extracts the features from the k-space signals, and generates the k-space tokens which are then concatenated with the instruction prompt to generate the image and the MRI analysis tokens. The image decoder takes the image tokens generated by the Nirvana backbone as the input and outputs the reconstructed MRI images. The standard U-Net architecture is employed to form the image decoder. To avoid the negative effect of the small quantity of k-space signals and improve the stability of the training process, we apply layer normalization before the k-space encoder and the image decoder.

During post-training for MRI reconstruction, the language backbone of the 1.3B Nirvana model remains frozen, while training is applied solely to the k-space encoder and image decoder. The post-training process comprises two sequential stages. In the first stage, we only train the k-space encoder guided by the Cross-Entropy (CE) loss of only the generated MRI analysis tokens. After the k-space encoder is trained until convergence, we freeze the k-space encoder as well as the Nirvana backbone, such that the training in the second stage does not influence Nirvana's performance on MRI analysis. We then only train the image decoder with the MRI image reconstruction loss. Following (Giannakopoulos et al., 2024; Jatyani et al., 2025), the model minimizes the Structural Similarity Index Measure (SSIM) (Sriram et al., 2020) loss between the reconstructed image $\hat{\mathbf{x}}$ and the ground truth image $\mathbf{x}^*$ in the second stage:

$$\mathcal{L}_2\left(\hat{\mathbf{x}}, \mathbf{x}^*\right) = -\operatorname{SSIM}\left(\hat{\mathbf{x}}, \mathbf{x}^*\right). \tag{9}$$

To evaluate the performance of Nirvana for MRI reconstruction, we use the FastMRI dataset (Zbontar et al., 2018). FastMRI dataset provides paired k-space signals and MRI images that can be directly used in the second post-training stage. In the first post-training stage, we create a list of possible instruction prompts, such as "According to the k-space signals, are there any pathological features?" The ground truth analysis of the MRI images corresponding to the instruction prompt is generated by the Lingshu Model (Xu et al., 2025).

MRI is greatly limited by a slow data acquisition process, which sometimes requires patients to remain still for an hour (Chen et al., 2022; Singh et al., 2023). Thus, it is essential to accelerate the MRI scan by undersampling in the scanning process. Following (Zbontar et al., 2018; Giannakopoulos et al., 2024; Jatyani et al., 2025), we undersample the k-space signals in the frequency domain to accelerate MRI acquisition while reducing the amount of data to be processed. Detailed undersampling configurations are provided in Table 10 of Appendix A.6.

We compare the performance of Nirvana for MRI reconstruction with other baselines under different undersampling rates in Figure 2. The MRI reconstruction performances of all models degrade when the undersampling rate becomes larger, because less information is provided in the higher-rate undersampled k-space signals. Nirvana surpasses the other models under all undersampling rates, and the Nirvana's performance degradation trend is the least significant as the undersampling rate becomes

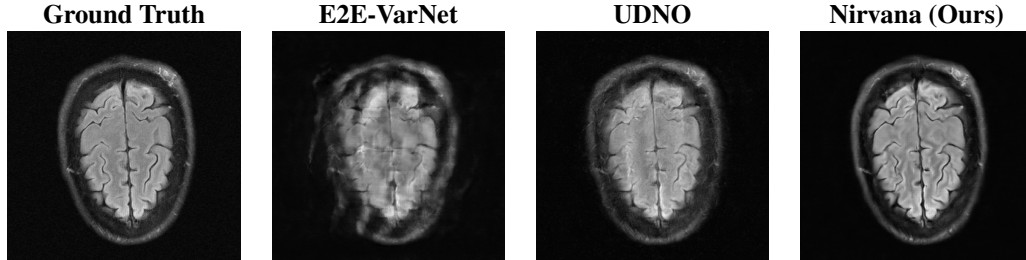

**Ground Truth**  **E2E-VarNet**  **UDNO**  **Nirvana (Ours)**

Figure 3: MRI reconstruction performance comparison for models with 160M trainable parameters. The acceleration rate, i.e., the undersampling rate, is set as 8 in the test time.

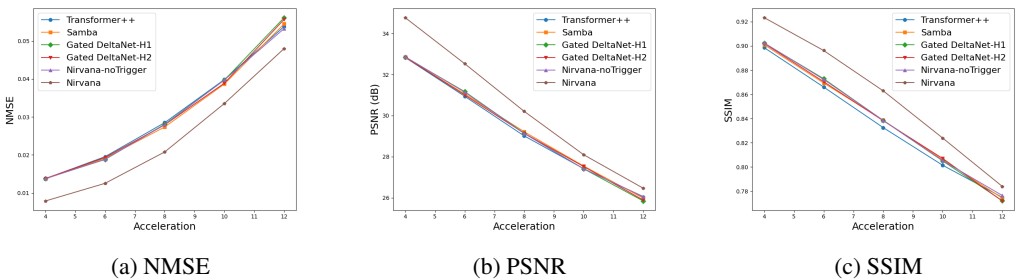

(a) NMSE  (b) PSNR  (c) SSIM

Figure 4: MRI reconstruction performance comparison between Nirvana and conventional LLMs with 160M trainable parameters in the k-space encoder and the image decoder. The acceleration rate is equivalent to the undersampling rate.

larger. This illustrates the potential advantage of Nirvana, which can use highly undersampled k-space signals to reconstruct the image while maintaining the same or even better image quality compared to E2E-VarNet and UDNO. Therefore, Nirvana has the potential ability to accelerate the scanning process of MRI.

We further visualize Nirvana's MRI reconstruction performance at an undersampling rate of 8 in Figure 3. The ground truth, the images reconstructed by E2E-VarNet, UDNO, and Nirvana are shown in the 4 columns, respectively. As shown in Figure 3, the performance of Nirvana is better than UDNO and E2E-VarNet in terms of the image fidelity and resolution. The reconstructed image of E2E-VarNet is blurry, and some part of the brain is completely obscured by black patches. The reconstructed image of UDNO is roughly close to the ground truth image, but the resolution is low and the details of the image are unclear. However, the reconstructed image of Nirvana is clear and accurate with high resolution, remarkably resembling the ground truth image with the highest SSIM, whose value is 0.8812.

We compare the MRI reconstruction performance of 1.3B Nirvana with other 1.3B LLMs across different undersampling rates in Figure 4. All models are pretrained on FineWeb and post-trained with the frozen backbone using the same procedure described at the beginning of Section 3.2.2. Nirvana consistently achieves higher reconstruction fidelity across all settings. In the ablation study, Nirvana-noTrigger shows substantially degraded performance, highlighting the critical role of Trigger in enabling effective adaptation for MRI reconstruction.

We show an example of the overall MRI reconstruction and report generation by Nirvana in Figure 5. Nirvana takes the undersampled k-space signals and the instruction prompt as input, and outputs the reconstructed MRI image as well as the corresponding analysis. In contrast with traditional MRI report generation models (such as Lingshu (Xu et al., 2025), HealthGPT (Lin et al., 2025), and MedGemma (Sellergren et al., 2025)) which directly take the reconstructed MRI images as input, Nirvana takes the k-space signals undersampled from the raw signals received by the coils as input to generate the overall MRI report, including the reconstructed image. In Figure 5, the report generated by Nirvana accurately captures the important pathological features of the image, including the lesion's shape, size, position, and surrounding matter. Moreover, the report provides further diagnosis that

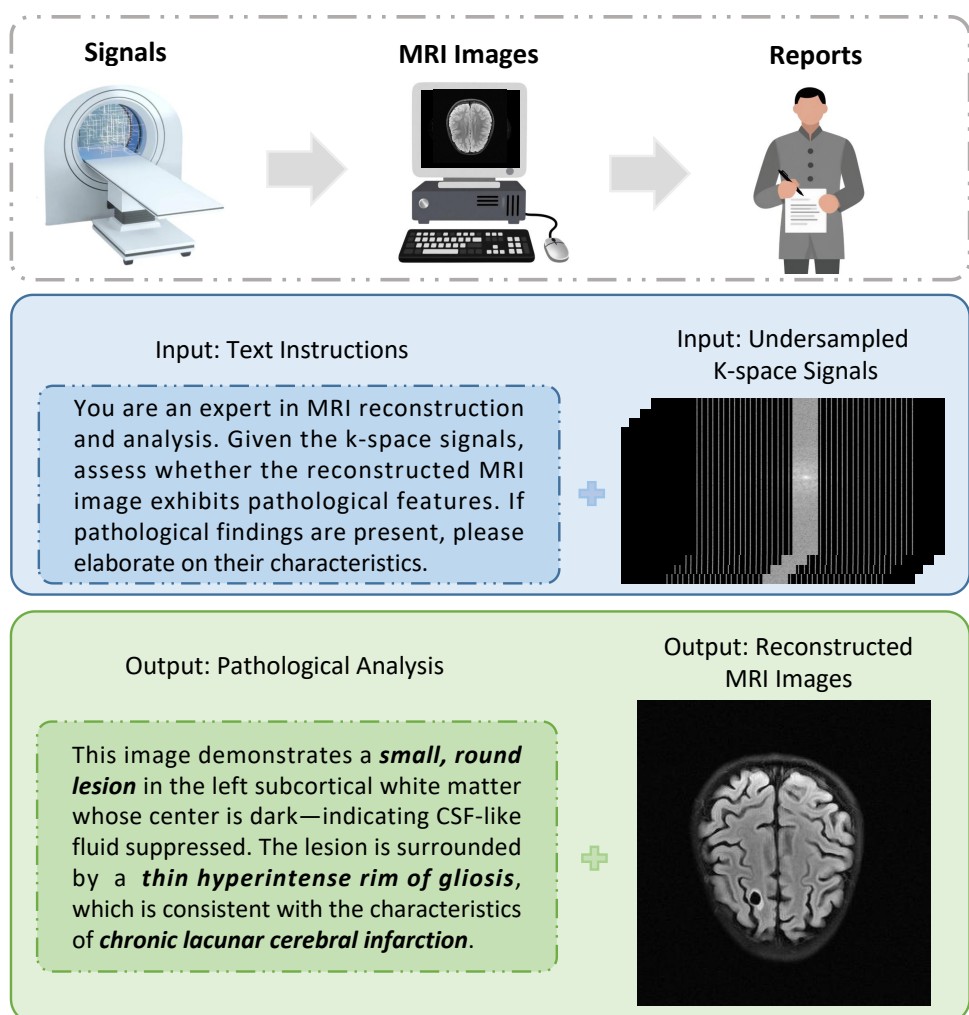

Figure 5: The overall process of MRI reconstruction and report generation by Nirvana.

the corresponding image is consistent with the characteristics of chronic lacunar cerebral infarction. More experiment results and analysis of MRI are shown in Appendix A.6.

## 4 CONCLUSION

In this work, we present Nirvana, an SGM with the task-aware memory mechanism. By enabling dynamic interpolation between SWA and Linear Attention, Updater allows the model to flexibly balance local and global information flow while maintaining computational efficiency. Complementing this, Trigger introduces per-sample self-supervision, allowing Nirvana to adapt to distributional shifts without requiring backbone retraining. Experiments show that Nirvana matches or exceeds strong LLM baselines on general benchmarks, and furthermore achieves the lowest perplexity across specialized domains including biomedicine, finance, and law. In the challenging MRI task, Nirvana yields higher-fidelity MRI reconstructions than conventional MRI models and LLM-based models, and it also generates reliable preliminary clinical reports. Importantly, ablation studies reveal that removing Trigger leads to notable performance degradation across all evaluation tasks, demonstrating its essential role in task-aware specialization. These findings indicate that Nirvana can transition smoothly from general language understanding to diverse specialized and high-precision domains.

## REPRODUCIBILITY STATEMENT

In this paper, we present a novel SGM called Nirvana. To guarantee that our work can be easily reproduced and built upon by the research community, we have taken the following key steps. First, the source code implementing our method is available as part of the supplementary materials. The code includes all scripts necessary for training and evaluating Nirvana, while pretrained models will be released after the reviewing process. Experimental settings and hyperparameters are available in our experiments and supplementary materials. We use publicly available datasets for training and evaluation, and details are reported in the experiments. Finally, we also provide information about the hardware environment used in our experiments.

We provide open-source code to reproduce our experiments at the following anonymous repository: https://anonymous.4open.science/r/Nirvana-SGM.

## ETHICS STATEMENT

This paper complies with the ethical guidelines of ICLR 2026. Our research has been conducted with a clear commitment to avoiding harm and upholding honesty and transparency in both methodology and reporting. We have made deliberate efforts to identify and mitigate potential biases in data and algorithms to promote fairness. In addition, we have respected individual privacy and ensured full compliance with all relevant regulations and ethical standards governing data use.

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

## A  APPENDIX

### A.1  STATEMENT FOR USE OF LLMS

LLMs were only used to assist with language polishing in certain sections of this paper.

### A.2  RELATED WORK

#### A.2.1  HYBRID ATTENTION-RECURRENT ARCHITECTURES

**Samba**  Samba (Ren et al., 2024) interleaves a simple state-space model (Mamba) with SWA to achieve effectively unbounded context lengths while retaining parallel training. It demonstrates strong performance on long-context language modeling tasks but relies on a fixed alternation pattern that may not suit all inputs.

**Jamba**  Jamba (Lieber et al., 2024) proposes a hybrid Transformer-Mamba architecture augmented with an MoE gating mechanism. By selectively activating Mamba or Transformer blocks, it scales to 256K-token contexts with high throughput. However, its static gating rules and layer placements require extensive tuning for each deployment scenario.

**Jamba-1.5**  Jamba-1.5 (Team et al., 2024) extends Jamba by scaling to instruction-tuned conversational models (12B and 94B parameters) and introducing 8-bit quantized experts (ExpertsInt8) for efficient inference. While quantization reduces memory footprint, the overall system complexity and reliance on large MoE layers can hinder adoption in resource-constrained settings.

**Gated DeltaNet (H1 & H2)**  Gated DeltaNet (Yang et al., 2025) builds on DeltaNet by employing learnable gates and Delta-rule updates within recurrent layers. The hybrid architectures combined with SWA, Mamba, and Gated Delta-Rule update are referred to as Gated DeltaNet (H1 & H2) (Yang et al., 2025). These models yield improved length-extrapolation and reasoning capabilities over Mamba2, but the added gating logic increases per-step computation and may introduce latency.

While these hybrid architectures advance long-context efficiency, they generally fix the ratio or placement of attention versus recurrence and lack per-sample adaptability. In contrast, Nirvana employs a *conditional interpolation mechanism* according to the task characteristics, realizing a specialized memory mechanism that adapts to the task domain.

#### A.2.2  TTT AND META-LEARNING

**TTT**  A special RNN framework called TTT is introduced by (Sun et al., 2024) with expressive hidden states that enable the model to perform a self-supervised gradient update on each test example. This strategy significantly improves robustness to domain shift but adds inference-time computation and may require stability controls.

**Meta-Learning Foundations**  Meta-learning (Vanschoren, 2019; Vettoruzzo et al., 2024) can be categorized into metric-based, model-based, and optimization-based methods. The highlighted challenges of meta-learning lie in transferring fast adaptation to streaming or per-sample settings.

**Online Meta-Learning** Online Meta-Learning is proposed by (Finn et al., 2019), which continually updates a meta-learner's parameters via a Model-Agnostic Meta-Learning (MAML)-style regret-minimization process over streaming tasks. While effective for sequential adaptation, it does not directly address efficient self-supervised tuning at inference.

Building on these insights, Nirvana integrates a Trigger module that treats each incoming sample as a self-supervised fine-tuning task, enabling rapid on-the-fly adaptation with minimal overhead and improved stability under distributional shift.

### A.3 A Toy Example of Nirvana in Combinatorial Tasks

To illustrate the effectiveness of Nirvana model, we consider a toy example of combinatorial tasks, where the model is required to conduct common sense reasoning while retrieving the question from a haystack. As shown in Figure 6, the haystack contains a set of repeated useless information, such as "the sky is blue" and "the grass is green". The key information, i.e., the question, is "Where is the capital of Switzerland?" The model should be able to retrieve the useful information at the beginning of the haystack and then answer the question. The Nirvana model accurately distinguishes the useful question from the useless information and then answers the question correctly. However, both Transformer++ and Gated DeltaNet fail to find the question and are misled to repeat the useless message instead. This demonstrates the superior performance of the Nirvana model over Transformer++ and Gated DeltaNet in combinatorial tasks of common sense reasoning and key information retrieval in long sequences.

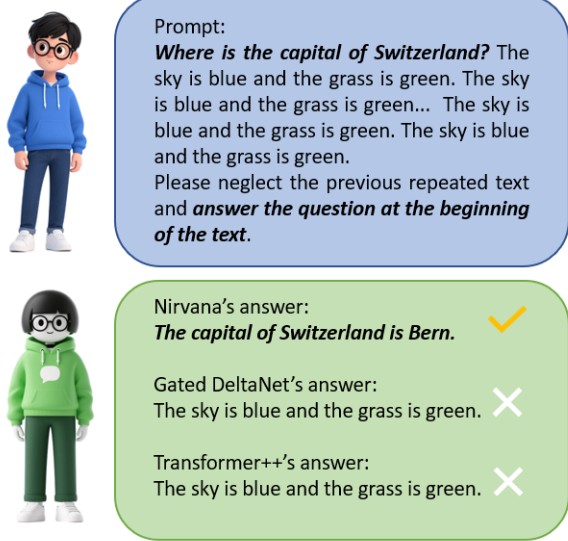

Figure 6: A toy example for combinatorial tasks of common sense reasoning and key information retrieval in long sequences.

### A.4 Experiments Related to RoPE

We investigate whether adding RoPE (Su et al., 2024) in SWA enhances the model's capability or not, where Nirvana-RoPE refers to the Nirvana model with RoPE added to the query and key of SWA modules. We first conduct experiments on NIAH in Table 6. Note that if the SWA module is added with RoPE, Nirvana-RoPE will be drastically worse than Nirvana with 8K context length, with accuracy of only 0.2% on S-NIAH-PK and only 4.4% on S-NIAH-N. This illustrates the importance of removing RoPE in SWA modules, which can lead to degraded performance when the context length at the test time is larger than that at the training time.

Table 6: S-NIAH performance of 1.3B models related to RoPE. S-NIAH-PK, S-NIAH-N, and S-NIAH-W are 3 tasks for the pass-key retrieval in a haystack, number in a haystack, and word in a haystack, respectively. All models are trained with 4K context length.

| Model | S-NIAH-PK | | | S-NIAH-N | | | S-NIAH-W | | | Average |
|---|---|---|---|---|---|---|---|---|---|---|
| | 2K | 4K | 8K | 2K | 4K | 8K | 1K | 2K | 4K | |
| Nirvana-RoPE | **100.0** | **100.0** | 0.2 | **100.0** | **100.0** | 4.4 | **100.0** | 97.0 | 92.8 | 77.2 |
| **Nirvana (Ours)** | **100.0** | **100.0** | **100.0** | **100.0** | **100.0** | **99.6** | 98.8 | **97.8** | **95.4** | **99.1** |

Table 7: Language Modeling and Zero-Shot Common Sense Reasoning Performance of 1.3B Models.

| Model | Wiki. | LMB. | LMB. | PIQA | Hella. | Wino. | ARC-e | ARC-c | SIQA | BoolQ | Avg. |
|---|---|---|---|---|---|---|---|---|---|---|---|
| | ppl ↓ | ppl ↓ | acc ↑ | acc ↑ | acc_n ↑ | acc ↑ | acc ↑ | acc_n ↑ | acc ↑ | acc ↑ | ↑ |
| Nirvana-RoPE | 18.01 | 12.13 | 49.97 | **73.71** | 58.17 | 58.43 | 68.90 | **38.86** | 41.15 | **59.33** | 56.07 |
| **Nirvana (Ours)** | **17.57** | **11.56** | **50.37** | 73.67 | **58.25** | **59.48** | **68.92** | 38.51 | **41.62** | 59.27 | **56.26** |

We conduct experiments on language modeling and zero-shot common sense reasoning in Table 7. The results illustrate that adding RoPE does not make Nirvana's performance better on average accuracy. The reason is that the Linear Attention architectures (e.g., Gated DeltaNet and Mamba2) are well qualified to capture the position-dependent information of the input sequence (Yang et al., 2025). Thus, there is no need to use RoPE in SWA, which would otherwise require additional computation and undermine the model's ability to extrapolate with context lengths longer than the training data.

## A.5 SUPPLEMENTARY LANGUAGE MODELING ABILITY

Table 8: Accuracy on 14 tasks from LongBench (Bai et al., 2023), including Narrative QA, QasperQA, MultiField QA, HotpotQA, 2WikiMulti QA, Musique, GovReport, QMSum, MultiNews, TRec, Trivia QA, SamSum, LCC, and RepoBench-P by order.

| Models | Single-Doc QA | | | Multi-Doc QA | | | Summarization | | | Few-shot | | | Code | | Avg |
|---|---|---|---|---|---|---|---|---|---|---|---|---|---|---|---|
| | NQA | QQA | MFQ | HQA | 2WM | Mus | GvR | QMS | MNs | TRC | TQA | SSM | LCC | RBP | |
| Recurrent models | | | | | | | | | | | | | | | |
| RetNet | 12.1 | 10.7 | 19.1 | 10.7 | 18.0 | 5.8 | 4.8 | 15.8 | 7.9 | 19.0 | 18.0 | 12.8 | 14.1 | 17.9 | 13.2 |
| HGRN2 | 10.7 | 12.1 | 19.1 | 11.3 | 15.7 | 6.0 | 5.2 | 15.1 | 9.2 | 16.0 | 15.8 | 10.3 | 18.6 | 20.8 | 13.5 |
| Mamba | 13.0 | 10.1 | 20.4 | 10.1 | 16.7 | 6.0 | 7.2 | 15.9 | 8.4 | 23.1 | 21.9 | 11.2 | 17.9 | 19.0 | 14.6 |
| DeltaNet | 12.9 | 10.8 | 21.5 | 10.9 | 13.2 | 5.1 | 6.5 | 13.5 | 7.2 | 15.5 | 23.3 | 11.6 | 17.6 | 20.3 | 13.6 |
| Mamba2 | 11.1 | 11.3 | 18.6 | 11.8 | 15.1 | 6.7 | 6.7 | 14.5 | 7.4 | 13.0 | 23.6 | 8.4 | 17.9 | 20.6 | 13.5 |
| Gated DeltaNet | 14.1 | 14.0 | 23.3 | 13.7 | 14.4 | 5.8 | 7.5 | 16.4 | 7.9 | 30.0 | 22.4 | 23.0 | 18.7 | 22.1 | 16.6 |
| Attention or hybrid models | | | | | | | | | | | | | | | |
| Transformer++ | 11.8 | 9.3 | 10.0 | 10.9 | 4.2 | 6.1 | 7.4 | 15.8 | 6.6 | 16.9 | 13.5 | 3.9 | 17.2 | 18.7 | 11.0 |
| Samba | 12.5 | 12.9 | 25.4 | 11.2 | 19.7 | 6.8 | 9.1 | 15.7 | 11.0 | 20.0 | 22.7 | 22.8 | 18.1 | 21.1 | 15.9 |
| Gated DeltaNet-H1 | 14.5 | 12.3 | 26.6 | 12.6 | 23.6 | 6.1 | 9.1 | 16.1 | 12.8 | 33.5 | 23.9 | 26.8 | 15.5 | 19.2 | 17.8 |
| Gated DeltaNet-H2 | 12.7 | **13.0** | 27.1 | 12.7 | 20.6 | 7.5 | **10.4** | **16.2** | 13.0 | **40.5** | 22.7 | **27.9** | **19.9** | **22.1** | 18.4 |
| Nirvana-noTrigger | 14.8 | 11.8 | 25.6 | 14.0 | 23.9 | 7.7 | 9.2 | 15.1 | 13.5 | 33.0 | 21.2 | 22.9 | 16.5 | 20.9 | 17.9 |
| **Nirvana (Ours)** | **16.6** | 12.8 | 26.0 | **14.6** | 24.8 | 9.7 | **10.4** | 15.9 | **15.4** | 36.4 | **25.2** | 22.6 | 17.5 | 21.5 | **19.2** |

We evaluate model performance on LongBench (Bai et al., 2023), a comprehensive suite of long-context tasks spanning retrieval, reasoning, multi-document understanding, and in-context learning. As shown in Table 8, Nirvana achieves consistent improvements across most categories—including NQA, HQA, 2WM, Mus, GvR, MNs, and TQA. These results highlight Nirvana's strengthened abilities in long-range retrieval, efficient in-context learning, and robust state tracking, demonstrating its effectiveness not only in general domains but also in specialized long-context understanding.

As shown in Fig. 7, we evaluate the models' capacity of extrapolating to sequences from 4K to 20K tokens across 3 long-context benchmarks, i.e., NarrativeQA, QMSum, and GovReport (Bai et al., 2023). Nirvana achieves the lowest overall perplexity across different tasks among all models. Besides, Nirvana without Trigger is also evaluated in Fig. 7, and its performance is not as good

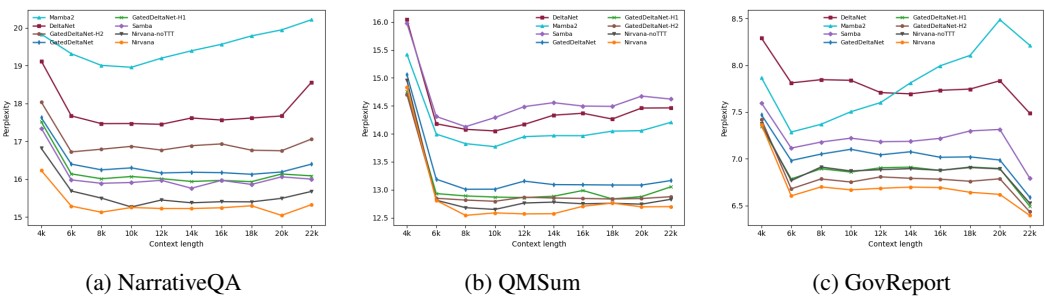

(a) NarrativeQA        (b) QMSum        (c) GovReport

Figure 7: Length extrapolation from 4K to 20K tokens on 3 long benchmarks.

as that of Nirvana. While we observe performance fluctuations when the context length becomes longer, Nirvana exhibits relatively more stable performance, which indicates that Nirvana is robust and has superiority in length-extrapolation tasks. We will explore Nirvana's capabilities on even longer sequences in the future.

Table 9: Accuracy on recall-world retrieval tasks with the input sequences truncated to 2K tokens, where SQD is short for SQUADE, and TQA is short for Trivial QA (Arora et al., 2024).

| Models | SWDE | SQD | FDA | TQA | NQ | Drop | Avg |
|---|---|---|---|---|---|---|---|
| Recurrent models | | | | | | | |
| RetNet | 14.0 | 28.5 | 7.0 | 54.4 | 16.2 | 17.3 | 22.9 |
| HGRN2 | 8.3 | 25.3 | 4.8 | 51.2 | 14.2 | 16.9 | 20.1 |
| Mamba | 9.8 | 25.8 | 3.7 | 54.3 | 14.9 | 17.4 | 21.0 |
| Mamba2 | 19.1 | 33.6 | 25.3 | 61.0 | 20.8 | 19.2 | 29.8 |
| DeltaNet | 17.9 | 30.9 | 18.4 | 53.9 | 17.3 | 18.6 | 26.2 |
| Gated DeltaNet | 25.4 | 34.8 | 23.7 | 60.0 | 20.0 | 19.8 | 30.6 |
| Attention or hybrid models | | | | | | | |
| Transformer++ | 29.5 | 38.0 | 52.2 | 58.3 | 22.5 | 21.6 | 37.0 |
| Samba | 33.0 | 39.2 | 50.5 | 57.7 | 23.5 | 20.2 | 37.3 |
| Gated DeltaNet-H1 | 35.6 | 39.7 | **52.0** | 60.1 | 24.6 | 22.2 | 39.0 |
| Gated DeltaNet-H2 | **38.2** | 40.4 | 50.7 | **63.3** | **24.8** | **23.3** | **40.1** |
| Nirvana-noTrigger | 35.1 | 39.8 | 50.5 | 60.0 | 22.3 | 21.7 | 38.2 |
| **Nirvana (Ours)** | 37.8 | **41.0** | 51.1 | 62.8 | **24.8** | 22.9 | **40.1** |

In Table 9, we present the models' accuracy on real-world recall-intensive tasks (Arora et al., 2024). Due to the limitations of linear attention, recurrent models show a significant performance gap compared to Transformers++, while Nirvana outperforms pure attention and achieves comparable performance with SOTA hybrid models in retrieval-intensive tasks. Without Trigger, Nirvana's performance will be notably degraded because of the lack of crucial task-aware memory management mechanism.

## A.6 SPECIALIZED ABILITY OF MRI RECONSTRUCTION

Table 10: The k-space undersampling configurations (acceleration and center fraction parameters) used for MRI reconstruction.

| Undersampling Rate | Acceleration Rate | Center Fraction Rate |
|---|---|---|
| 12× | 12 | 0.027 |
| 10× | 10 | 0.032 |
| 8× | 8 | 0.04 |
| 6× | 6 | 0.06 |
| 4× | 4 | 0.08 |

In MRI reconstruction, we undersample the k-space signals to accelerate the MRI coil scanning process in the frequency domain, and at the same time also reduce the amount of data to be processed (Zbontar et al., 2018; Giannakopoulos et al., 2024; Jatyani et al., 2025). The detailed k-space undersampling configurations are shown in Table 10.

Table 11: MRI reconstruction performance comparison of models with 160M trainable parameters. For Nirvana, the trainable components are the k-space encoder and the MRI decoder. The undersampling rate is set as 6 in this table during the test time.

| Model | SSIM ↑ | PSNR (dB) ↑ | NMSE ($\times 10^{-2}$) ↓ |
|---|---|---|---|
| E2E-VarNet | $0.8540 \pm 0.0418$ | $29.68 \pm 2.99$ | $2.512 \pm 0.742$ |
| UDNO | $0.8598 \pm 0.0414$ | $30.21 \pm 2.97$ | $2.074 \pm 0.730$ |
| **Nirvana (Ours)** | $\mathbf{0.9003} \pm 0.0407$ | $\mathbf{32.97} \pm 2.93$ | $\mathbf{1.176} \pm 0.625$ |

In Table 11, we evaluate the performance of Nirvana for MRI reconstruction using SSIM, PSNR, and NMSE, and also compare its performance with other baselines, including E2E-VarNet (Giannakopoulos et al., 2024) and UDNO (Jatyani et al., 2025). The undersampling rate is set as 6 in the test time. As shown in Table 11, Nirvana achieves the highest SSIM and PSNR, as well as the lowest NMSE on the test set. Besides, Nirvana's performance has the smallest variance and thus the highest stability. Specifically, Nirvana achieves an average improvement of 0.0405 in SSIM, 2.76 dB in PSNR, and $8.974 \times 10^{-3}$ in NMSE compared to the SOTA model UDNO (Jatyani et al., 2025), respectively.

