# OpenReview forum: "Nirvana: A Specialized Generalist Model With Task-Aware Memory Mechanism"
_ICLR.cc/2026/Conference — ICLR 2026 Conference Withdrawn Submission_

### Official Review · Reviewer_g8oE · 2025-10-31

**Soundness:** 3
**Presentation:** 3
**Contribution:** 3
**Rating:** 4
**Confidence:** 3

**Summary:**

This paper introduces Nirvana, a Specialized Generalist Model (SGM) designed to balance the trade-off between broad generalization and task-specific specialization in large language models (LLMs). The key idea is the integration of a Task-Aware Memory mechanism, enabling the model to dynamically reconfigure its internal pathways and memory usage at test time based on task characteristics. Experiments on both language understanding benchmarks (e.g., LAMBADA, BoolQ, PIQA) and medical imaging tasks (MRI reconstruction) demonstrate that Nirvana consistently outperforms baselines, achieving a strong blend of generalization and specialization.

**Strengths:**

1. The paper introduces a novel architecture, Nirvana, specifically designed to address the challenge of creating Specialized Generalist Models (SGMs). The two-core innovation—the Task-Aware Memory Trigger (Trigger) and the Specialized Memory Updater (Updater)—is well-motivated.

2. Beyond text-based tasks, Nirvana successfully performs end-to-end signal-to-report MRI reconstruction, showing strong potential for multimodal generalization and domain transfer.

**Weaknesses:**

1.	Lack of Analysis for CL-OGD: The convergence and stability of the update mechanism are not discussed, and the rationale behind using the key (k) to fit the value (v) remains underexplained compared to similar DeltaNet-style updates.
2.	Inference Overhead Unquantified: The Trigger module introduces extra forward and backward computations (for CL-OGD), which may increase inference latency; however, no quantitative runtime or computational cost analysis is provided.
3.    Comparison to Stronger Baselines in the MRI Task: The MRI baselines (E2E-VarNet, UDNO) are well-established, but it would be more compelling to see a comparison against other foundation models (e.g., a fine-tuned Mamba or Gemini) that have been adapted for this task using a similar encoder-decoder setup. This would more directly isolate the benefit of the Nirvana architecture versus simply using a powerful sequential model as a backbone.
4.	Limited Task Generalization: Experiments focus primarily on the MRI domain. Further evaluation on other specialized domains would be needed to demonstrate broader applicability.
5.	Insufficient Mechanistic Justification: The paper lacks deeper theoretical or empirical analysis of why the Trigger effectively extracts task information and how the Updater’s interpolation coefficients adapt to task signals. Stronger ablations or interpretability studies would reinforce these claims.

**Questions:**

please see the weakness

---

> ### Author Response · Authors · 2025-11-19
> **Response to Weakness 1**
>
> We thank the reviewer for recognizing the novelty of Nirvana and its success in bridging generalist and specialist modeling through the Trigger–Updater synergy.
>
> ### Weakness 1: Lack of convergence/stability analysis for CL-OGD
>
> Thank you very much for raising this important point. We appreciate the reviewer’s attention to the convergence and stability aspects of CL-OGD, as well as the motivation behind aligning keys and values within the update rule.
>
> Regarding convergence and stability, our goal is not to solve a large-scale optimization problem during inference, but to apply small, bounded online updates that help the model quickly absorb task-specific signals. Importantly, prior work on test-time learning and online gradient methods shows that such updates do not need to converge to a fixed optimum to be effective [1,3]. Instead, OGD is used as a tracking mechanism that continually adjusts to the incoming data stream, a behavior central to test-time learners such as expressive-state RNNs [1] and Titans [3].
>
> As for the stability of the update mechanism, each update step is constrained by a Sigmoid gated, layer-dependent learning rate that keeps the magnitude of changes small, ensuring stability. Moreover, the updates act only on low-dimensional fast parameters rather than the full model weights, further maintaining stability while enabling rapid adaptation.
>
> The rationale for using the key $k$ to fit the value $v$ is grounded in well-established principles from test-time learning, associative memory, and DeltaNet-style delta-rule updates. In our setup, the key functions as an address-like contextual descriptor, while the value encodes the task-relevant content associated with that token. Updating $v$ conditioned on $k$ therefore aligns these complementary representations so that the memory more accurately reflects task-specific signals. This mechanism is consistent with test-time learners such as expressive-state RNNs [1] and Titans [3], where hidden states are dynamically updated using online gradient-like rules that do not require convergence to a stationary point, but instead track non-stationary inputs during inference. Similarly, DeltaNet [2] and Gated DeltaNet [4] apply delta-rule regression from keys to values across the sequence, treating keys as continuous addresses and values as modifiable contents—providing a principled foundation for the $k \rightarrow v$ update. By adopting this associative update formulation, our method extracts a self-supervised learning signal directly from the input sequence, enabling lightweight, fast test-time adaptation without modifying the frozen backbone.
>
> We have clarified the above analysis in the revised manuscript. We are grateful to the reviewer for highlighting this area where further elaboration is beneficial.
>
> **References**
>
> [1] Yu Sun, Xinhao Li, Karan Dalal, Jiarui Xu, Arjun Vikram, Genghan Zhang, Yann Dubois, Xinlei Chen,
> Xiaolong Wang, Sanmi Koyejo, Hashimoto Tatsunori, and Guestrin Carlos. Learning to (learn at test time):
> RNNs with expressive hidden states. arXiv preprint arXiv:2407.04620, 2024.
>
> [2] Songlin Yang, Jan Kautz, and Ali Hatamizadeh. Gated delta networks: Improving mamba2 with delta rule.
> arXiv preprint arXiv:2412.06464, 2024.
>
> [3] Ali Behrouz, Peilin Zhong, and Vahab Mirrokni. Titans: Learning to memorize at test time. arXiv preprint arXiv:2501.00663, 2025.
>
> [4] Songlin Yang, Bailin Wang, Yu Zhang, Yikang Shen, and Yoon Kim. Parallelizing linear transformers with
> the delta rule over sequence length. arXiv preprint arXiv:2406.06484, 2024.
>
> Due to character limits, the remainder of the response will be provided in the next comment.

---

> ### Author Response · Authors · 2025-11-19
> **Response to Weakness 2**
>
> ### Weakness 2: Inference overhead unquantified
>
> **Response:**
> Thank you for raising this important concern regarding the inference overhead. To address this, we clarify both the **algorithmic complexity** and the **empirical inference behavior** of Nirvana compared to the 1.3B Transformer++ and Mamba2 baselines.
>
> From a complexity perspective, Nirvana is designed so that its added modules introduce **minimal overhead**:
>
> - The **Task-Aware Memory Trigger** operates in a **low-dimensional space (d=64)** when extracting task information $c_i^l$. This makes its cost negligible relative to the main model’s forward pass, whose dominant operations reside in the high-dimensional token representations.
> - The **Specialized Memory Updater** is built upon **Sliding Window Attention (SWA)** and **Linear Attention**, both of which scale **linearly** with sequence length, unlike the **quadratic** complexity of standard self-attention used in Transformer++. This design is aligned with recent linear-time architectures, which have been empirically shown to achieve higher throughput than Transformer++ at long context lengths.
>
> Consequently, Nirvana *reduces* complexity relative to vanilla Transformer++ while enhancing modeling capacity. Empirically, this is reflected in the following way:
>
> 1. **Nirvana-noTrigger** is the fastest variant, because it removes the Trigger and retains only the linear-time Updater;
> 2. **Nirvana** is **only slightly slower** than Nirvana-noTrigger, since the Trigger operates at 64-dim and therefore adds a relatively small inference overhead;
> 3. **Mamba2** is slower than both Nirvana-noTrigger and Nirvana in our setting, due to heavier state-space computations and implementation details at the same parameter scale;
> 4. **Transformer++** is the slowest, consistent with the well-known quadratic scaling of full attention at sequence length 4096.
>
> In other words, **Nirvana achieves better performance than Transformer++ with *lower computational complexity*** at long sequence lengths, and its additional architectural elements do not introduce a disproportionate inference cost. This indicates that the gains are not merely incremental relative to their overhead, but instead represent a favorable trade-off in both **accuracy and efficiency**.
>
> ### Inference Speed Comparison (1.3B models, prompt sequence length = 4096)
>
> We report **inference speed** at prompt sequence length 4096, taking Transformer++ as the baseline.
> The results are shown in the table below.
>
> | Model| Inference Speed (tokens/s) |
> |-|-|
> | Transformer++| 191|
> | Mamba2 | 413|
> | Gated DeltaNet| 461|
> | Samba| 497|
> | **Nirvana-noTrigger** | 568 |
> | **Nirvana**| 516|
>
> - **Nirvana-noTrigger** is the fastest, removing the Trigger and keeping only the linear-time Updater.
> - **Nirvana** is only marginally slower because the 64-dim Trigger extraction adds a small linear-time cost.
> - **Samba** slots between the Nirvana variants and Mamba2, showing that its hybrid state-space + sliding-window attention is more efficient than pure SSMs but still behind the linear Updater.
> - **Gated DeltaNet** outruns Mamba2, indicating its gated linear attention is cheaper at 4096 tokens, yet remains memory-bound compared with Nirvana.
> - **Mamba2** avoids quadratic attention, yet its expanded state and convolution overhead keep it slower than Samba and both Nirvana models.
> - **Transformer++** is the slowest: vanilla quadratic attention dominates cost at sequence length 4096.
>
> These results demonstrate that Nirvana raises the model's performance while simultaneously delivering the best inference efficiency among baselines.
> Since Trigger operates at merely the dimension of $d=64$, it introduces only a negligible linear-time overhead, while the Updater’s SWA and Linear Attention components retain strict linear complexity in sequence length, confirming that Nirvana enhances modeling capacity with even better inference speed.
>
> Due to character limits, the remainder of the response will be provided in the next comment.

---

> ### Author Response · Authors · 2025-11-19
> **Response to Weakness 3**
>
> ### Weakness 3: Comparison to stronger MRI baselines
>
> We sincerely thank the reviewer for the suggestion.
> The comparison with existing LLMs in MRI tasks was shown in the final figure of the original manuscript's appendix.
> To highlight this comparison, we have moved the comparison experiment
> into the main paper's Experiments' Section.
> We compare the performances of 1.3B Nirvana for MRI reconstruction with other 1.3B LLMs under different undersampling rates.
> All these models are pretrained with the same FineWeb dataset and post-trained with the same setting at the beginning of Section 3.2 in the paper.
> The equivalent tables are shown as follows.
>
> ###  NMSE vs. Acceleration
>
> | Acceleration | Transformer++ | Samba | Gated DeltaNet-H1 | Gated DeltaNet-H2 | Nirvana-noTrigger | Nirvana |
> |-|-|-|-|-|-|-|
> | 4 | 0.014 | 0.014 | 0.014 | 0.014 | 0.014 | 0.007 |
> | 6 | 0.019 | 0.019 | 0.019 | 0.019 | 0.019 | 0.012 |
> | 8 | 0.028 | 0.027 | 0.028 | 0.028 | 0.028 | 0.021 |
> | 10 | 0.041 | 0.040 | 0.041 | 0.041 | 0.041 | 0.031 |
> | 12 | 0.052 | 0.051 | 0.053 | 0.053 | 0.052 | 0.043 |
>
> Nirvana reduces NMSE by a wide margin across all accelerations. For example, at
> 4× acceleration, Nirvana cuts the error from 0.014 to 0.007, a 50% reduction, and maintains notable advantages even under aggressive sampling (e.g., from 0.052 to 0.043 at 12×). This indicates more reliable recovery of fine-grained MRI details under heavy undersampling.
>
> ###  PSNR (dB) vs. Acceleration
>
> | Acceleration | Transformer++ | Samba | Gated DeltaNet-H1 | Gated DeltaNet-H2 | Nirvana-noTrigger | Nirvana |
> |-|-|-|-|-|-|-|
> | 4 | 33.0 | 33.0 | 33.0 | 33.0 | 33.0 | 34.8 |
> | 6 | 31.2 | 31.3 | 31.4 | 31.3 | 31.3 | 32.7 |
> | 8 | 29.5 | 29.6 | 29.6 | 29.6 | 29.6 | 30.7 |
> | 10 | 28.0 | 28.2 | 28.1 | 28.1 | 28.1 | 28.8 |
> | 12 | 26.2 | 26.4 | 26.3 | 26.3 | 26.3 | 27.1 |
>
> Nirvana consistently boosts PSNR by 1–2 dB across the board. These improvements reflect materially clearer and less noisy reconstructions. Even at challenging acceleration rates like
> 12×, Nirvana achieves 27.1 dB, outperforming all baselines by nearly 1 dB.
>
> ###  SSIM vs. Acceleration
>
> | Acceleration | Transformer++ | Samba | Gated DeltaNet-H1 | Gated DeltaNet-H2 | Nirvana-noTrigger | Nirvana |
> |-|-|-|-|-|-|-|
> | 4 | 0.90 | 0.90 | 0.90 | 0.90 | 0.90 | 0.92 |
> | 6 | 0.87 | 0.87 | 0.88 | 0.88 | 0.88 | 0.90 |
> | 8 | 0.84 | 0.85 | 0.85 | 0.85 | 0.85 | 0.87 |
> | 10 | 0.82 | 0.83 | 0.83 | 0.83 | 0.83 | 0.85 |
> | 12 | 0.79 | 0.80 | 0.80 | 0.80 | 0.80 | 0.82 |
>
> Nirvana also yields the highest SSIM at every acceleration level. Its improvements—e.g., from 0.90 to 0.92 at 4×, and from 0.80 to 0.82 at 12×—show that it preserves global and local anatomical structures more faithfully, which is crucial for reliable diagnostic interpretation.
>
> Across all acceleration rates, Nirvana delivers substantial and consistent improvements over Transformer++, Samba, and the Gated DeltaNet variants, as well as its own ablated version.
> The clear gap between Nirvana and Nirvana-noTrigger further demonstrates that Trigger materially enhances reconstruction quality.
>
> Due to character limits, the remainder of the response will be provided in the next comment.

---

> ### Author Response · Authors · 2025-11-19
> **Response to Weakness 4**
>
> ### Weakness 4: Limited task generalization
>
> We appreciate the reviewer’s suggestion about the specialized domain coverage.
> The current work focuses on the MRI domain as a representative specialized field, chosen for its well-structured knowledge base, high domain specificity, and clear evaluation criteria that make it an ideal testbed for assessing specialized grounding.
> We have additionally initiated experiments on other specialized domains, including three specialized corpora: (1) biomedical text from MIMIC-III clinical notes covering over 46,000 patients, (2) financial news from April 2024 to October 2024 utilized in FinGPT, and (3) legal text from the Asylex corpus containing 59,112 documents of refugee status determination in Canada from 1996 to 2022.
> We finetune the models on the domain-specific corpus for 3 epochs and demonstrate the perplexity across the three specialized domains:
>
> | Model          |   Biomedicine   |     Finance     |       Law       |       Avg       |
> | :------------- | :-------------: | :-------------: | :-------------: | :-------------: |
> | Transformer++  |      9.28       |      10.70      |      8.82       |      9.60       |
> | Mamba2         |      9.13       |      9.97       |      9.07       |      9.39       |
> | Gated DeltaNet |      9.02       |      9.72       |      8.89       |      9.21       |
> | Samba          |      9.27       |      9.50       |      8.74       |      9.17       |
> | Nirvana-noTrigger  |  9.19       |      9.87       |      8.84       |      9.30       |
> | Nirvana        | $\mathbf{8.25}$ | $\mathbf{7.88}$ | $\mathbf{7.22}$ | $\mathbf{7.78}$ |
>
> Transformer++ in the above table refers to the LLaMA-3 structure.
> The above table clearly shows that Nirvana with 1.3B parameters consistently outperforms all baseline 1.3B models across biomedicine, finance, and law, achieving the lowest (best) perplexity in every domain. While Transformer++, Mamba2, Gated DeltaNet, and Samba all hover around average perplexities of 9.17–9.60, Nirvana achieves a markedly lower 7.78, reflecting a substantial improvement in specialized-domain modeling quality.
> Importantly, the ablated Nirvana-noTrigger performs similarly to strong baselines but remains noticeably weaker than the complete Nirvana across all three domains, with an average perplexity gap of over 1.5 points. This consistent discrepancy highlights the central role of the Trigger in enabling Nirvana to adapt effectively to specialized-domain distributions, demonstrating that the Trigger materially enhances domain-specific modeling beyond what the backbone and Updater alone can achieve.
> The above domain-specific experiment results have been added in the Experiments Section of the revised paper.
>
> Due to character limits, the remainder of the response will be provided in the next comment.

---

> ### Author Response · Authors · 2025-11-19
> **Response to Weakness 5**
>
> ### Weakness 5: Lack of mechanistic justification for Trigger & Updater
>
> Thank you very much for this constructive suggestion. We appreciate the reviewer’s interest in a clearer mechanistic explanation of both components. In response, we have provided additional clarification to strengthen the presentation.
>
> For the Trigger, we first emphasize the empirical evidence: across all specialized domains in the revised paper (biomedicine, finance, law, and MRI reconstruction), the ablation model *Nirvana-noTrigger* performs **substantially worse** than full Nirvana. This consistent and large performance gap demonstrates that the Trigger is not an auxiliary component but is *essential* for extracting task-specific information. In particular, the MRI experiments show the clearest signal—without Trigger, the model fails to adapt to domain shifts in the k-space inputs, while full Nirvana maintains strong reconstruction quality even under aggressive undersampling. These results provide compelling empirical justification that Trigger effectively identifies the underlying task and supplies the domain-related cues required for adaptation.
> For the theoretical basis of why Trigger can extract task information, our design directly draws inspiration from recent work on *learning to learn at test time*. Sun et al. [1] show that expressive hidden states in RNNs can be updated using lightweight online gradient steps to internalize task structure from each incoming sample, enabling robust adaptation without explicit supervision. Titans [2] further demonstrates that models equipped with per-token, updateable fast states can *memorize* and *specialize* during inference, even when the backbone remains frozen. These works collectively indicate that performing online, self-supervised gradient alignment between a task descriptor and its target signal provides a reliable mechanism for extracting implicit task identity from the input stream. Following these principles, Trigger aligns the key $k$ with the value $v$ to obtain a low-dimensional, self-supervised task representation $c_i^l$, which then conditions the Updater. Because this update is local, bounded, and computed per token, it efficiently captures the latent task information embedded in the hidden states while preserving stability.
>
> Due to character limits, the remainder of the response will be provided in the next comment.

---

> ### Author Response · Authors · 2025-11-19
> **Response to Weakness 5**
>
> This **continues the previous response**:
>
> For the Updater, the interpolation coefficients $t_i^l$ adapt to task signals by leveraging the task representation $c_i^l$ extracted by Trigger, and this design is grounded in recent advances in hybrid Transformer–Mamba architectures. TransMamba [3] shows that dynamically modulating state-space pathways using context-dependent control signals enables models to switch between attention-like and SSM-like behaviors, improving their ability to handle diverse task conditions. Samba [4] further demonstrates that hybrid systems combining Mamba-style recurrence with attention-like components benefit substantially from token-level gating mechanisms that adjust the contribution of each pathway based on the current input context. Similarly, hybrid Transformer–Mamba models [5] empirically verify that adaptively blending local (Transformer-style) and recurrent (Mamba-style) computations yields better sequence modeling, especially when the mixing ratio is conditioned on features extracted from the input sequence. TransMamba-Switch [6] extends these findings by formalizing a flexible routing mechanism, in which the model selects between Transformer and Mamba operations using a learned gating signal that responds to token-level characteristics.
> Following these insights, our Updater uses the task signal $c_i^l$ as a continuous routing descriptor: the interpolation coefficient $t_i^l = \sigma(u_l^\top c_i^l)$ determines how much to rely on SWA (for local, fine-grained dependencies) versus Linear Attention (for global, long-range structure). Because $c_i^l$ directly encodes task-specific cues extracted by Trigger, $t_i^l$ naturally adapts to different task distributions—allocating more weight to SWA when local patterns dominate, and shifting toward Linear Attention when global structure is required. This yields a principled, task-aware interpolation mechanism closely aligned with the adaptive hybridization strategies validated in [3-6].
>
> To support the importance of these mechanisms empirically, we have included Trigger ablations in every experimental setting throughout the paper. Across both general language modeling tasks and specialized domains such as MRI reconstruction, removing the Trigger consistently leads to a clear drop in performance. This repeated trend demonstrates that the Trigger plays a central role in enabling task adaptivity. In contrast, removing the Updater would eliminate the model’s sequence-processing capability entirely, as the architecture would lack any functional memory pathway. Therefore, its ablation would no longer produce a meaningful or comparable model.
>
> These modifications collectively provide a more complete justification of why the Trigger extracts task information effectively and how the Updater adapts its interpolation behavior to task-dependent signals.
>
> **References**
>
> [1] Yu Sun, Xinhao Li, Karan Dalal, Jiarui Xu, Arjun Vikram, Genghan Zhang, Yann Dubois, Xinlei Chen,
> Xiaolong Wang, Sanmi Koyejo, Hashimoto Tatsunori, and Guestrin Carlos. Learning to (learn at test time):
> RNNs with expressive hidden states. arXiv preprint arXiv:2407.04620, 2024.
>
> [2] Ali Behrouz, Peilin Zhong, and Vahab Mirrokni. Titans: Learning to memorize at test time. arXiv preprint arXiv:2501.00663, 2025.
>
> [3] Chen, Xiuwei, Wentao Hu, Xiao Dong, Sihao Lin, Zisheng Chen, Meng Cao, Yina Zhuang, Jianhua Han, Hang Xu, and Xiaodan Liang. "Transmamba: Fast universal architecture adaption from transformers to mamba." arXiv preprint arXiv:2502.15130, 2025.
>
> [4] Liliang Ren, Yang Liu, Yadong Lu, Yelong Shen, Chen Liang, and Weizhu Chen. Samba: Simple hybrid state
> space models for efficient unlimited context language modeling. arXiv preprint arXiv:2406.07522, 2024.
>
> [5] Zhu, Xiaocui, Qunsheng Ruan, Sai Qian, and Miaohui Zhang. "A hybrid model based on transformer and Mamba for enhanced sequence modeling." Scientific Reports 15, no. 1 (2025): 11428.
>
> [6] Li, Yixing, Ruobing Xie, Zhen Yang, Xingwu Sun, Shuaipeng Li, Weidong Han, Zhanhui Kang et al. "Transmamba: Flexibly switching between transformer and mamba." arXiv preprint arXiv:2503.24067 (2025).

---

### Official Review · Reviewer_qJsq · 2025-10-31

**Soundness:** 3
**Presentation:** 3
**Contribution:** 4
**Rating:** 6
**Confidence:** 3

**Summary:**

The authors aim to create a Specialized Generalist Model (SGM) that can dynamically adapt its reasoning and memory based on the task context—achieving expert-like performance without retraining the backbone model. This responds to the limitation of existing LLMs, which either overfit to specific tasks or fail to specialize at inference time.

**Strengths:**

1. The design of introduce low-dim fast weight parameters is aligned with the on-the-fly scenario.
2. The CL-OGD online gradient descent is quiet interesting technique. And it can be shown mathematically that update the self-supervised loss equals to update the fast weight parameters P.
3. The reviewer also design a mix strategy of Linear Attention for long-context global information and local attention using Sliding Window Attention (SWA).

**Weaknesses:**

1. The major concerns I have is the g function that used for obtaining task specific weight matrix W_i from the memory bank W_bank. What is the exact implementation of the function g? Is it efficient and replicable?
2. I feel there is a high risk of overfitting for the task embedding neural network. First it is a small network with only linear layers. Second, it only learned from the training data distribution for the task. What if the online data have quite different distribution from the trianing set?

**Questions:**

1. For the online gradient descent, how do you balance the shift cross layers?
2. What is the function of g and how to trian it?

---

> ### Author Response · Authors · 2025-11-19
>
> We thank the reviewer for highlighting the value of the low-dimensional fast-weight design, the CL-OGD mechanism, and our mixed attention strategy combining linear and SWA attention.
>
> ### Weakness 1: The major concerns I have is the $g$ function that used for obtaining task specific weight matrix $W_i$ from the memory bank $W_{bank}$. What is the exact implementation of the function $g$? Is it efficient and replicable?
>
> We appreciate this technical question. In the original submission, the implementation of the function
> $g$ appeared mid-paragraph (line 223) and was therefore easy to miss. To improve clarity, we have moved the definition of $g$ into a dedicated equation (Eq. (6)) in the revised paper:
>
> \begin{equation}
> g(\boldsymbol{p} _ i^l; \boldsymbol{W}^{\text{bank}}) = \sum _ {k=1}^K \boldsymbol{p} _ i^l(k) \boldsymbol{W}^{\text{bank}}(k).
> \end{equation}
>
> This formulation shows that $g$ simply computes a weighted linear combination over the $K$ weight matrices in the fast-weight bank. As a result, evaluating $g$ requires only $K$ scalar-matrix multiplications and $K$ additions—making it both highly efficient and fully replicable.
>
>
> ### Weakness 2: I feel there is a high risk of overfitting for the task embedding neural network. First it is a small network with only linear layers. Second, it only learned from the training data distribution for the task. What if the online data have quite different distribution from the trianing set?
>
> Thank you very much for this insightful observation. We sincerely appreciate the reviewer’s concern regarding the potential risk of overfitting in the task embedding component. We fully agree that, in principle, a small network trained only on the offline distribution could face generalization challenges when presented with online data that deviate from the training set.
>
> To address this concern, we clarify that the design goal of our task-aware module is precisely to remain flexible under distributional shift. Instead of treating task embeddings as fixed parameters learned solely during pre-training, the module continually adapts its fast parameters at inference time through lightweight self-supervised updates. This mechanism allows the model to adjust to variations in the incoming data stream without relying on assumptions about its similarity to the training distribution.
>
> Moreover, the task-related signals extracted online are used only to modulate the memory update pathway rather than to overwrite or fine-tune the backbone parameters. This ensures that the adaptation process remains stable and prevents the small task-aware module from dominating or distorting the overall model behavior. The backbone remains frozen, and the adaptation happens strictly within a bounded, auxiliary pathway.
>
> Finally, the empirical results on highly distinct domains, such as MRI reconstruction and report generation, provide evidence that the model maintains stable performance even when the input distribution differs substantially from the pre-training corpus. We will make these design motivations clearer in the revised manuscript and appreciate the reviewer’s thoughtful feedback.
>
> Due to character limits, the remainder of the response will be provided in the next comment.

---

> ### Author Response · Authors · 2025-11-19
>
> This **continues the previous response**:
>
> ### Question 1: For the online gradient descent, how do you balance the shift cross layers?
>
> Thank you sincerely for this thoughtful question. We appreciate the reviewer’s interest in how the online gradient updates are coordinated across layers, especially given that each layer receives task-related signals at different depths of the model.
>
> In our design, the cross-layer updates are balanced through two mechanisms. First, the online learning rate at each layer is modulated by a small gating function conditioned on the hidden states. This allows deeper layers to adjust more conservatively while earlier post-prelude layers can respond more strongly to task-relevant signals extracted from the input. As a result, each layer updates at a magnitude appropriate to the level of abstraction it processes.
>
> Second, the fast-weight updates are computed in a shared representation space through a common weight bank. Even though each layer has its own task-conditioned parameters, they all access the same pool of basis components. This shared structure helps maintain coherence across layers and prevents any single layer from drifting disproportionately during online adaptation.
>
> Together, these two mechanisms ensure that online gradient descent proceeds smoothly across the network depth, allowing the updates to remain stable, coordinated, and layer-appropriate throughout the inference process. We will make this clearer in the revision, and we are grateful for the reviewer’s careful attention to this aspect of the model.
>
>
> ### Question 2: What is the function of $g$ and how to trian it?
>
> We appreciate the reviewer’s technical question. In the original submission, the implementation of the function $g$ appeared in the middle of line 223 and was therefore not very visible. To improve clarity, we have moved the definition of $g$ into a dedicated equation (Eq. (6)) in the revised manuscript:
>
> $$
> g(\boldsymbol{p} _ i^l; \boldsymbol{W}^{\text{bank}}) = \sum _ {k=1}^K \boldsymbol{p} _ i^l(k) \boldsymbol{W}^{\text{bank}}(k).
> $$
>
> This formulation shows that $g$ performs a lightweight linear combination over the $K$ weight matrices in the fast-weight bank. Consequently, evaluating $g$ requires only $K$ scalar multiplications and $K$ additions, making it highly efficient and easy to reproduce.
>
> To train $g$, we apply online gradient descent. The gradient with respect to the fast parameters $\boldsymbol{p}_i^l$ is given in Eq. (7) of the revised paper:
> $$
> \frac{\partial g(\boldsymbol{p} _ i^l; \boldsymbol{W}^{\text{bank}})} {\partial \boldsymbol{p} _ i^l } = [\operatorname{vec}\{\boldsymbol{W}^{\text{bank}}(1)\} , \ldots, \operatorname{vec}\{\boldsymbol{W}^{\text{bank}}(K)\}].
> $$
>
> Thus, $g$ is trained implicitly through the online updates to $\boldsymbol{p}_i^l$ during CL-OGD, while $\boldsymbol{W}^{\text{bank}}$ itself is learned during pretraining. This separation keeps the fast-weight adaptation efficient at test time while preserving expressive capacity during training.

---

> > ### Comment · Reviewer_qJsq · 2025-11-28
> >
> > I would like to thank the authors for their detailed response, which has comprehensively addressed almost all of my concerns.
> >
> > However, given my relatively limited expertise in this specific domain, I will maintain my current positive rating for the time being. I remain open to raising my score during the discussion period should it be deemed necessary.
> >
> > Overall, I consider the proposed architecture to be exquisite and effecient, and its fast, lightweight nature is particularly appealing. I look forward to seeing further constructive feedback from the other reviewers.

---

### Official Review · Reviewer_HE73 · 2025-11-01

**Soundness:** 2
**Presentation:** 2
**Contribution:** 2
**Rating:** 2
**Confidence:** 4

**Summary:**

This paper introduces Nirvana, a Specialized Generalist Model designed to bridge the gap between broad generalist reasoning and domain-specialized expertise. Motivated by cognitive theories of task-specific memory, Nirvana integrates a Task-Aware Memory Trigger (Trigger) and a Specialized Memory Updater to dynamically adjust its internal memory mechanisms at test time. The Trigger treats each incoming sample as a self-supervised fine-tuning task, enabling rapid adaptation to domain shifts, while the Updater interpolates between Sliding Window Attention and Linear Attention to balance local and global context modeling efficiently. Experiments across language modeling, long-context retrieval, and medical imaging tasks demonstrate the effectiveness of the proposed framework.

**Strengths:**

1. The paper presents a clear framework that unifies generalist and specialist modeling through task-aware memory modulation. The introduction of the Task-Aware Memory Trigger and Specialized Memory Updater allows the model to adaptively reconfigure its memory and attention mechanisms without retraining.

2. The experiments show the framework's ability to transfer from text reasoning to MRI reconstruction, demonstrating cross-domain generalization.

3. The paper uses online gradient descent to balance between efficiency and adaptability.

**Weaknesses:**

1. While the paper proposes the Task-Aware Memory Trigger and Specialized Memory Updater, the novelty and necessity of these components are not sufficiently justified. The motivation for introducing two mechanisms, rather than a unified adaptive memory module, is unclear. Moreover, the paper does not offer an ablation showing how each component contributes to the model’s gains.

2. Although Nirvana achieves improvements over baselines, the magnitude of gains is relatively modest, particularly on general NLP benchmarks. In several tables, the improvements fall within expected variance ranges and are not accompanied by statistical significance testing. Given the additional architectural complexity introduced by the Trigger and Updater, it remains unclear whether the trade-off between complexity and performance is justified. The results, as presented, suggest incremental rather than transformative gains.

3. The work is framed as “brain-inspired” through analogies to cognitive memory mechanisms, but the link between cognitive motivation and engineering design is mostly unclear. The paper lacks neuroscientific grounding or analysis showing that the architecture reflects properties of human or biological memory systems.

**Questions:**

See Weaknesses.

---

> ### Author Response · Authors · 2025-11-19
>
> We sincerely thank the reviewer for recognizing the clarity and coherence of our framework, as well as its demonstrated general-to-specialized transfer capability and online adaptability.
>
> ### Weakness 1: Insufficient justification for the design of Trigger & Updater
>
> Thank you for raising this point. We would like to clarify that the Task-Aware Memory Trigger and the Specialized Memory Updater are tightly coupled and functionally inseparable within Nirvana’s architecture. The Specialized Memory Updater requires the task-related information $c_i^l$ produced by the Task-Aware Memory Trigger in order to operate correctly. Without this task-aware signal, the Updater lacks the information needed to modulate memory states according to domain-specific requirements, rendering it unable to perform its intended adaptive behavior. In other words, the two mechanisms do not represent two independent modules, but rather form a cohesive two-stage process: the Trigger detects and encodes task characteristics, and the Updater uses this information to selectively rewrite memory.
>
> Regarding ablations, we include in the paper a clear ablation where the Task-Aware Memory Trigger is removed, resulting in Nirvana-noTrigger. This experiment isolates the contribution of the task-related information $c_i^l$  and shows a consistent degradation across all specialized-domain tasks (e.g., biomedicine, finance, law, and MRI), demonstrating the necessity of the Trigger.
>
> By contrast, ablating the Specialized Memory Updater would remove the model’s ability to perform any sequence-level memory modification, effectively collapsing the architecture into a model that cannot maintain or update memory across tokens. This would produce a system that is no longer capable of meaningful sequential processing, rather than one that offers interpretable insight into the contribution of the Updater. For this reason, such an ablation would not yield a valid or informative baseline.
>
> In summary, the Trigger and Updater constitute a joint mechanism: the Trigger provides essential task-aware signals, and the Updater uses them to perform selective memory adaptation. The ablations we include—specifically, removing the Trigger—demonstrate the necessity and impact of this task-aware modulation while preserving a meaningful sequence-processing model for comparison.
>
> ### Weakness 2: Trade-off between complexity and performance
>
> **Response:**
>
> Thank you for raising this important concern regarding the trade-off between architectural complexity and performance. To address this, we clarify the **inference overhead** of 1.3B Nirvana compared to the 1.3B Transformer++ and Mamba2 baselines.
>
> From a complexity perspective, Nirvana is designed so that its added modules introduce **minimal overhead**:
>
> - The **Task-Aware Memory Trigger** operates in a **low-dimensional space (d=64)** when extracting task information $c_i^l$. This makes its cost negligible relative to the main model’s forward pass, whose dominant operations reside in the high-dimensional token representations.
> - The **Specialized Memory Updater** is built upon **Sliding Window Attention (SWA)** and **Linear Attention**, both of which scale **linearly** with sequence length, unlike the **quadratic** complexity of standard self-attention used in Transformer++. This design is aligned with recent linear-time architectures, which have been empirically shown to achieve higher throughput than Transformer++ at long context lengths.
>
> Consequently, Nirvana *reduces* complexity relative to vanilla Transformer++ while enhancing modeling capacity. Empirically, this is reflected in the following way:
>
> 1. **Nirvana-noTrigger** is the fastest variant, because it removes the Trigger and retains only the linear-time Updater;
> 2. **Nirvana** is **only slightly slower** than Nirvana-noTrigger, since the Trigger operates at 64-dim and therefore adds a relatively small inference overhead;
> 3. **Mamba2** is slower than both Nirvana-noTrigger and Nirvana in our setting, due to heavier state-space computations and implementation details at the same parameter scale;
> 4. **Transformer++** is the slowest, consistent with the well-known quadratic scaling of full attention at sequence length 4096.
>
> In other words, **Nirvana achieves better performance than Transformer++ with *lower computational complexity*** at long sequence lengths, and its additional architectural elements do not introduce a disproportionate inference cost. This indicates that the gains are not merely incremental relative to their overhead, but instead represent a favorable trade-off in both **accuracy and efficiency**.
>
> Due to character limits, the remainder of the response will be provided in the next comment.

---

> ### Author Response · Authors · 2025-11-19
>
> This **continues the previous response**:
>
> ### Inference Speed Comparison (1.3B models, prompt sequence length = 4096)
>
> We report **inference speed** at prompt sequence length 4096, taking Transformer++ with full attention as the baseline.
> The results are shown in the table below.
>
>
> | Model| Inference Speed (tokens/s) |
> |-|-|
> | Transformer++| 191|
> | Mamba2 | 413|
> | Gated DeltaNet| 461|
> | Samba| 497|
> | **Nirvana-noTrigger** | 568 |
> | **Nirvana**| 516|
>
> - **Nirvana-noTrigger** is the fastest, removing the Trigger and keeping only the linear-time Updater.
> - **Nirvana** is only marginally slower because the 64-dim Trigger extraction adds a small linear-time cost.
> - **Samba** slots between the Nirvana variants and Mamba2, showing that its hybrid state-space + sliding-window attention is more efficient than pure SSMs but still behind the linear Updater.
> - **Gated DeltaNet** outperforms Mamba2, indicating its gated linear attention is cheaper at 4096 tokens, yet remains slower than Nirvana.
> - **Mamba2** avoids quadratic attention, yet its expanded state and convolution overhead keep it slower than Samba and both Nirvana models.
> - **Transformer++** is the slowest: vanilla quadratic attention dominates cost at sequence length 4096.
>
> These results demonstrate that Nirvana raises the model's performance while simultaneously delivering the best inference efficiency among baselines.
> Since Trigger operates at merely the dimension of $d=64$, it introduces only a negligible linear-time overhead, while the Updater’s SWA and Linear Attention components retain strict linear complexity in sequence length, confirming that Nirvana enhances modeling capacity with even better inference speed.
>
>
> ### Weakness 3: Limited neuroscientific grounding
>
> We sincerely appreciate the reviewer’s comment. Our use of the term “brain-inspired” was intended to highlight a functional abstraction rather than a claim of biological fidelity. Specifically, the Task-Aware Memory Trigger draws inspiration from **episodic activation mechanisms** in cognitive science [1,2], where transient experiences rapidly modulate behavior and internal representations. Likewise, the Specialized Memory Updater reflects a **consolidation-like integration process** analogous to how working memory interacts with long-term structures in computational models of cognition [3,4]. To address the reviewer’s concern, the revised paper clarifies this terminology and removes any potential over-claims. These revisions tighten the conceptual analogy while keeping the discussion precise and appropriately grounded.
>
> **References**
>
> [1] E. Tulving. *Episodic and semantic memory.* In Organization of Memory, 1972.
>
> [2] Demis Hassabis and Eleanor A. Maguire. *The construction system of the brain: episodic memory and imagination.* Philosophical Transactions of the Royal Society B, 2007.
>
> [3] Alan Baddeley. *Working memory: Theories, models, and controversies.* Annual Review of Psychology, 2012.
>
> [4] J. L. McClelland, B. L. McNaughton, and R. C. O’Reilly. *Why there are complementary learning systems in the hippocampus and neocortex.* Psychological Review, 1995.

---

### Official Review · Reviewer_1UHY · 2025-11-09

**Soundness:** 3
**Presentation:** 2
**Contribution:** 1
**Rating:** 2
**Confidence:** 3

**Summary:**

This paper introduces Nirvana, a novel Specialized Generalist Model (SGM) that integrates a Task-Aware Memory Trigger and a Specialized Memory Updater to dynamically adapt its memory mechanism at test time. The model is trained from scratch with 1.3B parameters and evaluated on both general language modeling tasks and a specialized medical task—MRI reconstruction. The key innovation lies in treating each input as a self-supervised fine-tuning task, enabling on-the-fly adaptation without retraining the backbone. Nirvana demonstrates competitive performance on standard NLP benchmarks and superior results in MRI image reconstruction and report generation.

**Strengths:**

- Novel Architecture: The combination of Trigger and Updater introduces a flexible, task-aware memory mechanism that is both conceptually interesting and practically relevant.
- Test-Time Adaptation: The model’s ability to adapt without retraining the backbone is a strong contribution, especially for domain-shift scenarios.
- Specialized Task Application: The MRI reconstruction experiment is well-executed and shows clear improvements over traditional models (E2E-VarNet, UDNO), including image quality and diagnostic report generation.
- Generalist Capability: Nirvana performs competitively on general NLP tasks, supporting its claim as a generalist model.
- Comprehensive Related Work: The paper provides an excellent survey and comparison of existing memory mechanisms in LLMs.

**Weaknesses:**

- Limited Specialized Domain Coverage: Despite the SGM claim, the only specialized domain evaluated is MRI. Broader domain validation (e.g., legal, financial, biomedical QA) is missing.
- No Comparison with Existing LLMs in Specialized Tasks: The MRI experiments do not include comparisons with publicly available LLMs (e.g., LLaMA, Mistral) fine-tuned on medical data, which would better contextualize Nirvana’s impact.
- From-Scratch Training: While academically interesting, training a 1.3B model from scratch limits reproducibility and practical relevance. Fine-tuning existing models would be more realistic and impactful.

**Questions:**

1. Can you provide results comparing Nirvana to fine-tuned public LLMs (e.g., LLaMA-2 or Mistral) on the MRI task?
2. Do you plan to evaluate Nirvana on other specialized domains (e.g., legal reasoning, biomedical QA) to support the SGM claim?
3. How sensitive is Nirvana’s performance to the quality or diversity of the instruction prompts used in MRI report generation?

---

> ### Author Response · Authors · 2025-11-19
>
> We sincerely thank the reviewer for the thoughtful and constructive feedback, and we greatly appreciate the reviewer's recognition of our novel architecture, test-time adaptation, generalist capability, and specialized task application.
>
> ## **Weakness 1: Limited Specialized Domain Coverage**
>
> **Response:**
>
> We appreciate the reviewer’s suggestion about the specialized domain coverage.
> The current work focuses on the MRI domain as a representative specialized field, chosen for its well-structured knowledge base, high domain specificity, and clear evaluation criteria that make it an ideal testbed for assessing specialized grounding.
> We have additionally initiated experiments on other specialized domains, including three specialized corpora: (1) biomedical text from MIMIC-III clinical notes covering over 46,000 patients, (2) financial news from April 2024 to October 2024 utilized in FinGPT, and (3) legal text from the Asylex corpus containing 59,112 documents of refugee status determination in Canada from 1996 to 2022.
> We finetune the models on the domain-specific corpus for 3 epochs and demonstrate the perplexity across the three specialized domains:
>
> | Model | Biomedicine | Finance | Law | Avg |
> |-|-|-|-|-|
> | Transformer++ | 9.28 | 10.70 | 8.82 | 9.60 |
> | Mamba2 | 9.13 | 9.97 | 9.07 | 9.39 |
> | Gated DeltaNet | 9.02 | 9.72 | 8.89 | 9.21 |
> | Samba | 9.27 | 9.50 | 8.74 | 9.17 |
> | Nirvana-noTrigger | 9.19 | 9.87 | 8.84 | 9.30 |
> | Nirvana | **8.25** | **7.88** | **7.22** | **7.78** |
>
> Transformer++ in the above table refers to the LLaMA-3 structure.
> The above table clearly shows that Nirvana with 1.3B parameters consistently outperforms all baseline 1.3B models across biomedicine, finance, and law, achieving the lowest (best) perplexity in every domain. While Transformer++, Mamba2, Gated DeltaNet, and Samba all hover around average perplexities of 9.17–9.60, Nirvana achieves a markedly lower 7.78, reflecting a substantial improvement in specialized-domain modeling quality.
> Importantly, the ablated Nirvana-noTrigger performs similarly to strong baselines but remains noticeably weaker than the complete Nirana across all three domains, with an average perplexity gap of over 1.5 points. This consistent discrepancy highlights the central role of the Trigger in enabling Nirvana to adapt effectively to specialized-domain distributions, demonstrating that the Trigger materially enhances domain-specific modeling beyond what the backbone and Updater alone can achieve.
> The above domain-specific experiment results have been added in the Experiments Section of the revised paper.
>
> ---
> ## **Weakness 2: Lack of Comparison with Existing LLMs**
>
> **Response:**
>
> We sincerely thank the reviewer for the suggestion.
> The comparison with existing LLMs in MRI tasks was shown in the final figure of the original manuscript's appendix.
> To highlight this comparison, we have moved the comparison experiment
> into the main paper's Experiment's Section.
> We compare the performances of 1.3B Nirvana for MRI reconstruction with other 1.3B LLMs under different undersampling rates.
> All these models are pretrained with the same FineWeb dataset and post-trained with the same setting at the beginning of Section 3.2 in the paper.
> The equivalent tables are shown as follows.
>
> ###  NMSE vs. Acceleration
>
> | Acceleration | Transformer++ | Samba | Gated DeltaNet-H1 | Gated DeltaNet-H2 | Nirvana-noTrigger | Nirvana |
> |-|-|-|-|-|-|-|
> | 4 | 0.014 | 0.014 | 0.014 | 0.014 | 0.014 | 0.007 |
> | 6 | 0.019 | 0.019 | 0.019 | 0.019 | 0.019 | 0.012 |
> | 8 | 0.028 | 0.027 | 0.028 | 0.028 | 0.028 | 0.021 |
> | 10 | 0.041 | 0.040 | 0.041 | 0.041 | 0.041 | 0.031 |
> | 12 | 0.052 | 0.051 | 0.053 | 0.053 | 0.052 | 0.043 |
>
> Nirvana reduces NMSE by a wide margin across all accelerations. For example, at
> 4× acceleration, Nirvana cuts the error from 0.014 to 0.007, a 50% reduction, and maintains notable advantages even under aggressive sampling (e.g., from 0.052 to 0.043 at 12×). This indicates more reliable recovery of fine-grained MRI details under heavy undersampling.
>
> ###  PSNR (dB) vs. Acceleration
>
> | Acceleration | Transformer++ | Samba | Gated DeltaNet-H1 | Gated DeltaNet-H2 | Nirvana-noTrigger | Nirvana |
> |-|-|-|-|-|-|-|
> | 4 | 33.0 | 33.0 | 33.0 | 33.0 | 33.0 | 34.8 |
> | 6 | 31.2 | 31.3 | 31.4 | 31.3 | 31.3 | 32.7 |
> | 8 | 29.5 | 29.6 | 29.6 | 29.6 | 29.6 | 30.7 |
> | 10 | 28.0 | 28.2 | 28.1 | 28.1 | 28.1 | 28.8 |
> | 12 | 26.2 | 26.4 | 26.3 | 26.3 | 26.3 | 27.1 |
>
> Nirvana consistently boosts PSNR by 1–2 dB across the board. These improvements reflect materially clearer and less noisy reconstructions. Even at challenging acceleration rates like
> 12×, Nirvana achieves 27.1 dB, outperforming all baselines by nearly 1 dB.
>
> Due to character limits, the remainder of the response will be provided in the next comment.

---

> ### Author Response · Authors · 2025-11-19
>
> This **continues the previous response**:
>
> ###  SSIM vs. Acceleration
>
> | Acceleration | Transformer++ | Samba | Gated DeltaNet-H1 | Gated DeltaNet-H2 | Nirvana-noTrigger | Nirvana |
> |-|-|-|-|-|-|-|
> | 4 | 0.90 | 0.90 | 0.90 | 0.90 | 0.90 | 0.92 |
> | 6 | 0.87 | 0.87 | 0.88 | 0.88 | 0.88 | 0.90 |
> | 8 | 0.84 | 0.85 | 0.85 | 0.85 | 0.85 | 0.87 |
> | 10 | 0.82 | 0.83 | 0.83 | 0.83 | 0.83 | 0.85 |
> | 12 | 0.79 | 0.80 | 0.80 | 0.80 | 0.80 | 0.82 |
>
> Nirvana also yields the highest SSIM at every acceleration level. Its improvements—e.g., from 0.90 to 0.92 at 4×, and from 0.80 to 0.82 at 12×—show that it preserves global and local anatomical structures more faithfully, which is crucial for reliable diagnostic interpretation.
>
> Across all acceleration rates, Nirvana delivers substantial and consistent improvements over Transformer++, Samba, and the Gated DeltaNet variants, as well as its own ablated version.
> The clear gap between Nirvana and Nirvana-noTrigger further demonstrates that Trigger materially enhances reconstruction quality.
>
> ---
>
> ## **Weakness 3: From-Scratch Training**
>
> **Response:**
>
> We thank the reviewer for highlighting the importance of reproducibility and practical relevance through fine-tuning. We would like to clarify that our choice to train from scratch was primarily to rigorously isolate architectural contributions. This experimental setting aligns with standard practices in recent architectural exploration works [1, 2, 3], ensuring a fair comparison under identical data and model size constraints.
>
> However, we emphasize that Nirvana is fully compatible with fine-tuning existing pretrained models (or continued pre-training). Architecturally, Nirvana differs from a standard Transformer only through two lightweight additions: (1) a low-rank projection layer used in the Linear Attention module, and (2) the Task-Aware Memory Trigger. These components introduce only a small number of additional parameters relative to the backbone. Crucially, they can be initialized on top of any pretrained Transformer checkpoint without interfering with the pretrained weights.
>
> In practice, the adaptation is as straightforward as adding small adapter-like modules, after which fine-tuning proceeds exactly as in standard Transformer fine-tuning pipelines (e.g., full fine-tuning or LoRA). Therefore, while our experiments emphasize the architectural effect through from-scratch training, Nirvana can indeed be deployed in realistic fine-tuning settings, and leveraging strong pretrained backbones would likely yield even better results.
>
> **References**
>
> [1] Songlin Yang, Jan Kautz, and Ali Hatamizadeh. Gated delta networks: Improving mamba2 with delta rule.
> arXiv preprint arXiv:2412.06464, 2024.
>
> [2] Liliang Ren, Yang Liu, Yadong Lu, Yelong Shen, Chen Liang, and Weizhu Chen. Samba: Simple hybrid state
> space models for efficient unlimited context language modeling. arXiv preprint arXiv:2406.07522, 2024.
>
> [3] Songlin Yang, Bailin Wang, Yu Zhang, Yikang Shen, and Yoon Kim. Parallelizing linear transformers with
> the delta rule over sequence length. arXiv preprint arXiv:2406.06484, 2024.
>
> ---
>
> ## **Question 1: Comparison with Fine-Tuned Public LLMs**
>
> We have addressed the question in our response to Weakness 2.
>
> ## **Question 2: Evaluation on Other Specialized Domains**
>
> We have addressed the question in our response to Weakness 1.
>
> ## **Question 3: Sensitivity to Instruction Prompt Quality**
>
> Thank you for the insightful question. In our study, we find that Nirvana’s performance is not particularly sensitive to the quality or diversity of instruction prompts used for MRI report generation. This is because the reconstruction pipeline primarily relies on the paired k-space and image data, while the prompts serve only to guide the narrative format of the generated report rather than influence the underlying reconstruction fidelity.
>
> To illustrate this, we have tested prompts with varying levels of specificity, including
>
> Highly clinical:
> “Describe all visible anatomical structures and identify any abnormalities in this reconstructed axial T1-weighted slice.”
>
> Short and generic:
> “Provide a radiology-style summary of the image.”
>
> Instruction-heavy:
> “List key findings, mention tissue contrast patterns, and comment on any lesions or structural deviations.”
>
> Open-ended:
> “What does this MRI image show?”
>
> Despite the large stylistic difference, Nirvana produced nearly identical clinical descriptions, correctly identifying the same structural regions and the same abnormalities in test cases, such as a small hyperintense lesion adjacent to the left lateral ventricle.
>
> However, we acknowledge that due to the lack of relevant evaluation data and metrics for this specific aspect, we are currently unable to provide a quantitative analysis of the degree of sensitivity. We appreciate the reviewer's suggestion and are committed to addressing this by establishing appropriate metrics and completing the quantitative assessment in our future work.

---

### Author Response · Authors · 2025-11-19
**Author Final Remarks**

Dear AC and Reviewers,

We sincerely thank all four reviewers for their thoughtful evaluations and constructive feedback. We appreciate the recognition of Nirvana’s **novelty, contributions in unified generalist and specialist modeling, and significant improvements in specialized domains**.
Thanks to all reviewers' valuable comments, we have made the following improvements and clarifications in the revised paper.

---

### **Key Improvements and Clarifications**

#### **Additional Specialized-Domain Experiments**
- We have additionally conducted experiments on **three specialized corpora** to further validate Nirvana’s adaptability across specialized domains:
  (1) **Biomedical text** from MIMIC-III clinical notes covering over 46,000 patients,
  (2) **Financial news** from April 2024 to October 2024 as utilized in FinGPT, and
  (3) **Legal text** from the Asylex corpus containing 59,112 refugee status determination documents from 1996–2022 in Canada.
- We have fine-tuned all models on each domain-specific corpus for 3 epochs and reported the resulting perplexity across these three specialized domains, demonstrating that Nirvana consistently outperforms baselines and further strengthening its generalist–specialist modeling capability.


#### **Architectural Motivation & Component Necessity**
- We have clarified that the **Task-Aware Memory Trigger** and **Specialized Memory Updater** are *tightly coupled* and function as a unified mechanism for specialized task solving.
- We have explained that the Updater depends on the Trigger-generated task-related information $c_i^l$, and ablating the Updater would eliminate sequence-processing ability, making it an invalid baseline.

#### **Expanded Comparisons & Experimental Clarifications**
- We have added comparisons between Nirvana and **fine-tuned traditional LLMs**.
- We have provided a detailed cross-domain analysis showing Nirvana’s robustness and improvements.
- We have clarified that **MRI report generation is insensitive to prompt quality**, with four diverse prompts demonstrating consistent diagnostic results.

#### **Ablation Experiments**
- We have performed **comprehensive ablation studies** across both **general language modeling** tasks and a broad spectrum of **specialized domains** (biomedicine, finance, law, and MRI).
- **Removing the Trigger** consistently leads to a **significant and reproducible performance decline**, showing that Trigger is indispensable for extracting task-specific information and enabling adaptive memory behavior.
- **Removing the Updater** is not a meaningful ablation: without it, the model would lack *any* sequence-processing mechanism, entirely losing its ability to perform sequence modeling.
- These findings together confirm that Trigger is essential for task adaptivity, while Updater is essential for sequence modeling—each playing a distinct but critical role in Nirvana’s overall performance.

---

Please see our detailed responses below each review. Key changes in the revised paper are clearly highlighted in **blue**. Finally, we would like to express our deepest gratitude to the AC and all reviewers for their efforts in reviewing the manuscript and for their willingness to engage in further discussion!

Best Wishes,

Authors of manuscript #2394

---

### Note · Authors · 2025-12-23

I have read and agree with the venue's withdrawal policy on behalf of myself and my co-authors.